# APC2 controls dendrite development by promoting microtubule dynamics

Olga I. Kahn[1], Philipp Schätzle[1], Dieudonnée van de Willige[1], Roderick P. Tas[1], Feline W. Lindhout[1], Sybren Portegies[1], Lukas C. Kapitein[1] & Casper C. Hoogenraad[1]

Mixed polarity microtubule organization is the signature characteristic of vertebrate dendrites. Oppositely oriented microtubules form the basis for selective cargo trafficking in neurons, however the mechanisms that establish and maintain this organization are unclear. Here, we show that APC2, the brain-specific homolog of tumor-suppressor protein adenomatous polyposis coli (APC), promotes dynamics of minus-end-out microtubules in dendrites. We found that APC2 localizes as distinct clusters along microtubule bundles in dendrites and that this localization is driven by LC8-binding and two separate microtubule-interacting domains. Depletion of APC2 reduces the plus end dynamics of minus-end-out oriented microtubules, increases microtubule sliding, and causes defects in dendritic morphology. We propose a model in which APC2 regulates dendrite development by promoting dynamics of minus-end-out microtubules.

[1] Cell Biology, Department of Biology, Faculty of Science, Utrecht University, 3584 CH Utrecht, The Netherlands. Correspondence and requests for materials should be addressed to C.C.H. (email: c.hoogenraad@uu.nl)

Dendritic arborization gives rise to one of the most functionally and morphologically complex structures in the nervous system. Dendrites rely on microtubules for functional and structural support. Microtubules are intrinsically polar, dynamic polymers that form the foundation for directed motor-based transport of various cargoes, such as mitochondria, RNA particles and synaptic vesicles[1–3]. In vertebrate dendrites, microtubules adopt mixed orientations with roughly half oriented with their plus ends toward the cell body[1–5]. Microtubule motors recognize the orientation of microtubules and preferentially move either towards their minus (e.g. cytoplasmic dynein) or plus ends (e.g., kinesin-1)[6–8]. In this manner, cargo carried by dynein can enter dendrites, but not axons in which nearly all microtubules are organized uniformly plus-end-out. In invertebrate neurons, several mechanisms have been shown to play a role in microtubule organization, including local microtubule nucleation and motor-based transport of microtubules[3]. For example, the Golgi apparatus can locally nucleate microtubules with preferred respective polarity in axons and dendrites[9–12], while motors and microtubule-associated proteins can stabilize microtubules with proper orientations[13–18]. Nevertheless, the mechanism to organize and maintain mixed polarity microtubule orientations in vertebrate dendrites is largely unclear.

The brain specific homolog of tumor-suppressor protein adenomatous polyposis coli, adenomatous polyposis coli 2 (APC2), is known to regulate the neuronal cytoskeleton[19–21]. In Drosophila neurons, APC2 is localized to branch points in dendrites and directs microtubule growth in complex with APC, kinesin-2 and EB1 to maintain uniform minus-end-out microtubule polarity specific to Drosophila dendrites[22,23]. In vertebrate brains, APC2 plays a role in neuronal migration and axon guidance via stabilization of microtubules and regulation of actin dynamics through activation of Rho family GTPases[20,24,25]. APC2-deficient mice develop severe cortical layering flaws due to defects in directional migration in response to extracellular guidance molecules[21]. Recently, APC2 was reported to contribute to the development of Sotos syndrome[25–27]. Sotos syndrome is a type of cerebral gigantism, characterized by varying degrees of mental retardation and increased head size accompanied by characteristic facial features[26,27]. The neural features of the syndrome arise from either downregulation of APC2 controlled by NSD1 or from a homozygous frameshift mutation in the APC2 gene, where a single-nucleotide duplication results in premature truncation of the C-terminus microtubule binding domain.

Here, we describe a role for mammalian APC2 in regulation of dendritic microtubule dynamics. We find that during neuronal development, APC2 localizes in dendrites in distinct clusters along microtubules. This localization is determined by LC8 (DYNLL2) binding, +TIP interactions and the microtubule-binding domain within the C-terminus tail. Depletion of APC2 leads to a reduction of minus-end-out microtubule dynamics, resulting in decreased arborization of the dendritic tree. We propose a model in which APC2 regulates dendrite development by anchoring along newly polymerized microtubules and establishing potential rescue points, thereby increasing dynamicity of the dendritic microtubule cytoskeleton.

## Results

### Dendrite development is impaired in neurons depleted of APC2. 
In chick retinal axons, APC2 depletion has been shown to increase branching, destabilize microtubules and lead to axonal curving during outgrowth[20]. While axons contain a large population of stable microtubules, dendrites rely more on microtubule dynamics for their structural and functional integrity[28–33]. In order to determine the role of APC2 during late neuronal development, we depleted endogenous protein from cultured rat hippocampal neurons at two different time points. We targeted neurons from DIV7 to DIV11 to select for early dendrite development and from DIV14 to DIV18 to select for later dendrite and dendritic spine development (Fig. 1a). The efficiency of our shRNA was quantified by qPCR on cultured rat hippocampal neurons (Fig. 1b, Supplementary Fig. 1a). APC2 depletion did not result in significant changes in dendrite thickness (Fig. 1c), however depletions during both early and late stages lead to fewer primary dendrites (Fig. 1d) and simplified dendritic arbors as determined by Sholl analysis (Fig. 1e,f). At the end of late stage depletion, at DIV18, we quantified changes in density and morphology of dendritic spines (Fig. 1g–i). We found that control and APC2 depleted neurons contained similar levels of functional dendritic spine markers Homer and Bassoon (see legend for Fig. 1g). However, APC2-depleted neurons contained fewer spine protrusions per 50 μm stretches of dendrite (Fig. 1h). This decrease in density was due to fewer mushroom-headed spines (protrusions longer than 2 μm), filopodia spines (thin protrusions longer than 1 μm) and stubby spines (protrusions shorter than 1 μm) (Fig. 1i). Together, these results indicate that APC2 is necessary for proper development of the dendritic arbor and its maturation.

### APC2 clusters localize to MT bundles in dendrites.
APC2 is preferentially expressed in the nervous system from early developmental stages through adulthood. In early developmental stages APC2 is distributed along microtubules and actin in growth cones, as well as axon shafts of chick retinal axons and cerebellar granule cells[20,21,25]. Later in development, in Drosophila central neurons and dendritic arborization neurons, APC2 is localized to cell bodies, dendrites, and some proximal axons, but not distal axons[34]. The C-terminus of APC2 is vital for its proper localization and function. In patients with Sotos syndrome, a frameshift mutation in the APC2 gene, resulting in loss of the C-terminus, drives development of neurological symptoms[25]. We, therefore, focused on RFP-tagged full length human APC2 (RFP-APC2), APC2 without the C-terminus (RFP-APC2-ΔC) and only the C-terminus (RFP-APC2-C) for localization studies (Fig. 2a). To understand the mechanism by which APC2 affects dendritic morphology in vertebrate neurons, we first examined its localization throughout development (Fig. 2b). Early in neuronal development (DIV3), RFP-APC2 localizes with slight preference to the longest neurite, while in mature neurons RFP-APC2 overwhelmingly localizes to dendrites (DIV7 and DIV15). In mature neurons, both RFP-APC2 and RFP-APC2-ΔC localize exclusively to dendrites and stop their entry into the axon at the axon initial segment, indicating that restriction of the protein to dendrites at that stage is driven by the N-terminus. RFP-APC2-C bound all along the microtubules and quickly deformed neuronal morphology (Fig. 2c). Importantly, localization of RFP-APC2 in mammalian neurons is distinct from its localization in Drosophila dendrites, where it is constrained to branch points and acts together with kinesin-2, EB1 and APC to direct microtubule growth to achieve uniform minus-end-out microtubule polarity specific to Drosophila dendrites[22,23]. In mammalian dendrites, RFP-APC2 does not accumulate at branch points, but localizes in a regular cluster pattern all throughout the length of the dendrites (Fig. 2d). Localization of RFP-APC2 clusters closely resembles endogenous localization of APC2 in human iPSC-derived neurons (Supplementary Fig. 1b).

In mature neurons, RFP-APC2 clusters colocalize with both tyrosinated (freshly polymerized) and acetylated (longer-lived) microtubule bundles (Fig. 2e). Colocalization of RFP-APC2 clusters with microtubules was measured in cell bodies of

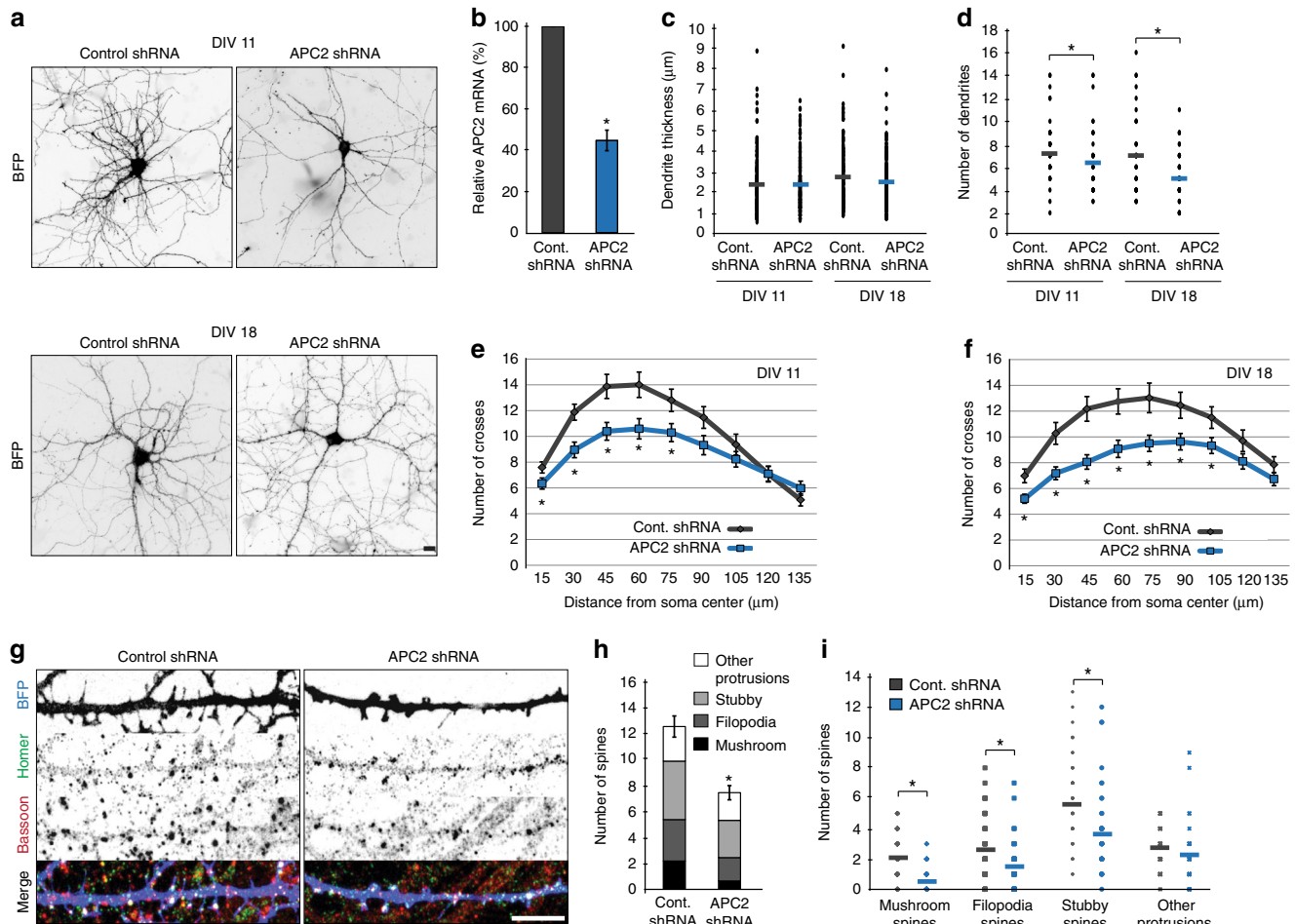

**Fig. 1** Dendrite development is impaired in neurons depleted of APC2. **a** Representative images of dendritic morphology of neurons transfected with MARCKS-BFP for cell outline and either control or APC2 shRNA with MARCKS-BFP. **b** mRNA levels in hippocampal neurons after 3 days of either control or APC2 shRNA expression. Data normalized to GAPDH ($n = 3$, $p = 0.008$, One sample $t$-test). **c** Dendrite thickness was measured within 5 µm of the cell body ($n = 188$ dendrites, $p = 0.963$ DIV11, $p = 0.206$ DIV18, $N = 2$, Mann–Whitney U test). **d** Number of primary dendrites was quantified per cell body ($n = 39$ neurons control shRNA DIV11, $n = 41$ neurons APC2 shRNA DIV11, $p = 0.009$, $n = 40$ neurons DIV18, $p = 0.020$, $N = 2$, Mann–Whitney U test). **e** Sholl analysis of dendritic branching at DIV11 ($n = 39$ neurons control shRNA, $n = 41$ neurons APC2 shRNA, $N = 2$, Two-Way ANOVA, Tukey post-hoc). **f** Sholl analysis of dendritic branching at DIV18 ($n = 40$ neurons, $N = 2$, Two-Way ANOVA, Tukey post-hoc). **g** Representative images at DIV18 of dendritic spines in control or APC2 shRNA with MARCKS-BFP transfected neurons. Homer and Bassoon levels in dendritic spines between control and APC2 depleted neurons did not differ. ($n = 42$ spines control shRNA, $n = 37$ spines APC2 shRNA; Homer: control shRNA 6797 ± 470 fluor. units vs. APC2 shRNA 8360 ± 783 fluor. units, $p = 0.125$, Mann–Whitney U test; Bassoon: control shRNA 16670 ± 923 fluor. units vs. APC2 shRNA 16441 ± 1348 fluor. units, $p = 0.589$, $N = 2$, Mann–Whitney U test). **h** Total number of spines positive for Homer and Bassoon quantified per 50 µm length of dendrite at DIV18. ($n = 50$ dendrites control shRNA, $n = 42$ dendrites APC2 shRNA, $p \leq 0.001$, $N = 2$, Independent samples $t$-test). **i** Number of spines positive for Homer and Bassoon was classified and quantified per 50 µm length of dendrite at DIV18. Mushroom spines are > 2 µm long with mushroom shape ($p \leq 0.001$), filopodia are >1 µm with no thickening ($p = 0.001$) and stubby spines are ≤1 µm ($p \leq 0.001$). Other protrusions ($p = 0.091$) were those that were negative for Homer and/or Bassoon staining ($n = 50$ dendrites control shRNA, $n = 42$ dendrites APC2 shRNA, $N = 2$, Mann–Whitney U test). Graphs represent mean ± SEM. * $p \leq 0.05$. Scale bars are 10 µm in **a** and 5 µm in **g**

neurons, in which majority of unpolymerized protein was extracted prior to fixation. RFP-APC2 did not colocalize preferentially with one population of microtubules over the other (see legend Fig. 2e). This colocalization of APC2 with microtubules relies on its C-terminal domain, since APC2 lacking the C-terminus is functionally null and colocalizes with lysosome maker LAMP2[25]. We also find that the C-terminus of APC2 drives its cluster localization and immobilization in dendrites. RFP-APC2 clusters are immobile, while RFP-APC2-ΔC puncta are larger, resemble aggregate-like structures and often undergo transport-like movements (Fig. 2f, g).

Interestingly, not all RFP-APC2 clusters colocalize with microtubule bundles. APC2 has been shown to localize to filamentous actin in *Drosophila* S2 cells[24] and play a role in actin

nucleation and organization[35]. To determine the extent to which APC2 relies on actin or microtubules for its localization, we either treated RFP-APC2 transfected neurons with the microtubule-depolymerizing drug nocodazole or co-transfected them with actin disassembly inductor SpvB (genetically encoded DeAct[36]) (Supplementary Fig. 1c). APC2 has been shown to protect microtubules from nocodazole-driven destabilization[25], therefore we did not expect complete relocalization of RFP-APC2 clusters (Fig. 2h). However, we did find that nocodazole treatment both reduced fluorescence intensity of the clusters and decreased their number with larger clusters remaining immobile, suggesting that a significant population of RFP-APC2 is diffusing upon microtubule depolymerization (Fig. 2h, i). Perturbation of actin via expression of GFP-SpvB did not lead to significant changes to

RFP-ACP2 localization (Fig. 2j, Supplementary Fig. 1c), suggesting that RFP-ACP2 does not rely on an intact actin cytoskeleton for proper localization.

To assess RFP-APC2 interaction with the cytoskeleton in more depth, we evaluated the degree of RFP-APC2 turnover either post nocodazole treatment or in the presence of SpvB. To do this, we performed FRAP on dendrites of neurons transfected with RFP-APC2 alone, together with GFP-SpvB or post 90 min treatment with nocodazole (Fig. 2j). We found that perturbation of both the actin and microtubule cytoskeleton affected the

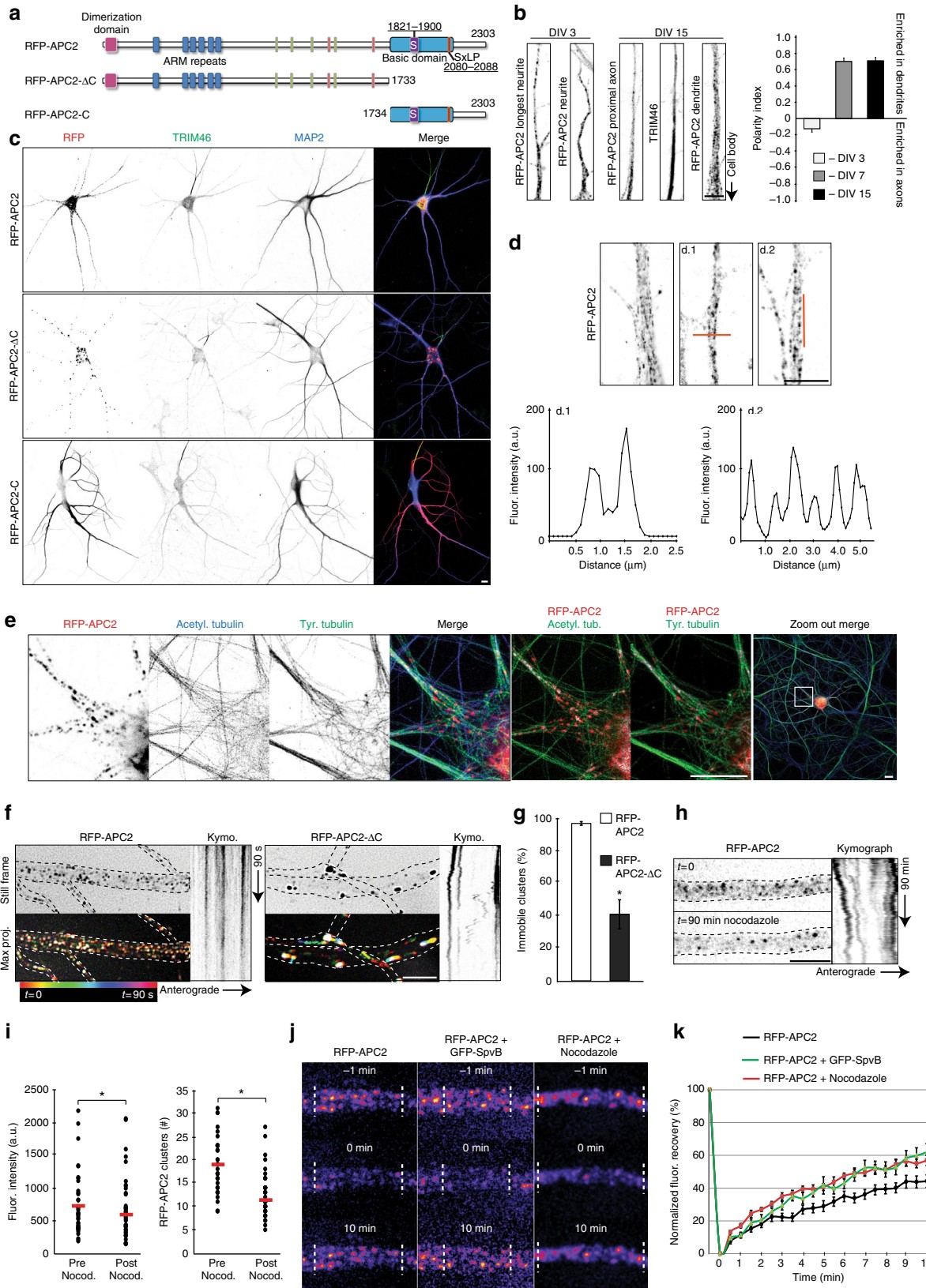

turnover of RFP-APC2 clusters in mammalian dendrites (Fig. 2h–k).

In order to determine if a known factor participates in the particular clustered localization of RFP-APC2, we immunostained neurons with known interactors of APC family proteins: Axin[37,38], β-catenin[39], and co-expressed N-cadherin[40]. None of the factors fully colocalized with RFP-APC2 and neither did microtubule minus-end protein CAMSAP2, Golgi protein GM130 or vesicle markers Rab6 and Rab11 (Supplementary Fig. 1d). Together these results indicate that RFP-APC2 clusters preferentially localize to microtubule bundles in dendrites via a mechanism that does not solely rely on partners from the Wnt signaling pathway.

**APC2 depletion alters minus-end-out MT dynamics in dendrites.** Since APC2 depletion resulted in significant dendritic morphological changes, we set out to determine its effect on microtubule dynamics and organization. Neurons were transfected at DIV11–12 with either control or APC2 shRNA together with GFP-MT + TIP and imaged at DIV15–16 to visualize orientation of dynamic microtubules (Fig. 3a). We quantified orientation and dynamics of microtubules from 3 min movies taken of proximal 6.7 µm stretches of primary dendrite (prior to 1st branch point) and 6.7 µm stretches of secondary dendrite (post 1st branch point). APC2 depletion resulted in a reduction of retrograde GFP-MT + TIP comets compared to control neurons (Fig. 3b, c). This effect was rescued by full length RFP-APC2, but not RFP-APC2-ΔC (Supplementary Fig. 2a). Curiously, the remaining dynamic minus-end-out microtubules only differed in their length of polymerization events, but not polymerization velocity from control, while plus-end-out microtubules underwent shorter and faster polymerization events in primary dendrites (Supplementary Fig. 2b, c). We found that the loss in minus-end-out microtubule dynamics due to APC2 depletion occurs primarily in the outer dendritic shaft (Fig. 3d–h). These results suggest that APC2 preferentially affects dynamics of a subset of microtubules in vertebrate dendrites and that its depletion results in a significant decrease in dynamics of those minus-end-out microtubules.

In addition, we found that APC2 plays a distinct role in mammalian dendrites compared to *Drosophila* dendrites[23]. Apart from no specific localization of RFP-APC2 to branch points (Fig. 2c, d) we found no APC2-dependent microtubule polymerization steering at branch points (Supplementary Fig. 2d–f). In order to quantify microtubule polymerizing plus ends using the innate steering mechanisms, we experimentally increased the number of microtubule polymerization events going through the branch point. To do this, we used photo-ablation to cut microtubules prior to and post branch point immediately preceding the movie recording. Cutting microtubule bundles with a short-pulsed laser creates new microtubule ends and allows analysis of the newly generated growing microtubule plus ends[5]. We classified steering of microtubules at branch points into five primary tracks. In *Drosophila*, primarily the red track (minus-end-out microtubule polymerizing into opposite periphery of dendritic shaft and sharply steered towards cell body) has been described to be guided by kinesin-2, APC, APC2 and EB1. Here, we show that APC2 depletion has no effect on sharply steered blue, green or red microtubule polymerization tracks (Supplementary Fig. 2e). These results highlight the distinct role of mammalian APC2 in dendritic microtubule dynamics.

**APC2 role in global organization of MT polarity in dendrites.** We next assessed whether the decrease in minus-end-out microtubule polymerization events in dendrites depleted of APC2 is due to a change in the number of oppositely oriented microtubules. To address global changes in cytoskeleton organization, we first checked for levels of microtubule sliding in the dendrites. Neurons were transfected with photoactivatable GFP-tubulin, which is dark until photoactivation, RFP-tubulin and either control or APC2 shRNA at DIV11–12 and imaged at DIV15–16. We photoactivated 5–10 µm long segments of proximal dendrites and imaged their behavior over 6 h (Fig. 4a). We noted several behaviors of the photoactivated region. First, a portion of the signal was lost due to depolymerization of photoactivated microtubules and likely repolymerization using RFP-tubulin. This loss of signal primarily happened in outer dendrite, in line with a higher number of dynamic microtubules present there[41]. Second, the photoactivated region displayed two different microtubule displacement behaviors. Full translocation of the converted region as a unit, which was always directed towards the soma (Supplementary Movie 1), and microtubule bundle sliding, where bundles slid out of the photoactivated region both toward and away from the soma (Fig. 4b). Distance of whole microtubule region translocation was significantly increased in APC2-depleted neurons (Fig. 4c). Of the control neurons, 31.0% displayed no movement of the region, 27.6% displayed microtubule bundle sliding and 41.4% displayed whole region translocation. In APC2-depleted neurons, 5.0% displayed no movement of the region, 35.0% displayed microtubule bundle sliding and 60.0% displayed whole region translocation. There was no difference in distance of microtubule bundle sliding between conditions (Fig. 4d). These results suggest that APC2 plays a role in interlocking the

---

**Fig. 2** APC2 clusters localize to microtubule bundles in dendrites. **a** Three constructs were made for these experiments based on human APC2 sequence. **b** Localization of RFP-APC2 was assessed throughout development of neurons at DIV3 ($n = 18$), DIV7 ($n = 19$) and DIV15 ($n = 15$) by measuring PI, $N = 2$. **c** Localization pattern of RFP-APC2, RFP-APC2-ΔC and RFP-APC2-C in DIV16 neurons immunostained for axon initial segment (TRIM46) and dendrites (MAP2). **d** RFP-APC2 localizes in a regular pattern clusters in primary dendrites. d.1 and d.2 are secondary dendrites, where RFP-APC2 often localizes to the outer dendrite. Line scans highlight the clustered pattern localization. **e** RFP-APC2 clusters colocalize with tyrosinated and acetylated microtubule bundles in neurons. Colocalized pixels are white, non-colocalized RFP-APC2 pixels are red and non-colocalized tubulin pixels are green ($n = 21$ areas, 0.42 ± 0,03 Manders' coefficient for acetylated tubulin and 0.39 ± 0.03 Manders' coefficient for tyrosinated tubulin, $p = 0.484$, Independent samples $t$-test). **f** Representative frames, color-coded maximum projections of 180 frames and kymographs of RFP-APC2 or RFP-APC2-ΔC in dendrites at DIV15–16. **g** Immobilized clusters were quantified as a percentage of total over 3 min movies taken every 0.5 s in 6.7 µm long stretches of dendrite. ($n = 31$ dendrites RFP-APC2, $n = 13$ dendrites RFP-APC2-ΔC, $p \leq 0.001$, $N = 4$, Mann–Whitney U test). **h** Neurons transfected with RFP-APC2 were imaged on DIV16 after treatment with 10 µM nocodazole. Kymograph is of 90 min movie with frames taken every 1 min. **i** Fluorescence intensity of RFP-APC2 was quantified pre and post nocodazole treatment in 10 µm same areas of dendrite after bleach correction ($n = 40$ dendrites, $p = 0.006$, Wilcoxon signed-rank test). Number of RFP-APC2 clusters was quantified pre and post nocodazole treatment in 10 µm areas of dendrite ($n = 40$ dendrites, $p \leq 0.001$, N = 4, Wilcoxon signed-rank test). **j** Representative images of DIV15–16 neurons transfected with RFP-APC2 and either GFP-SpvB or after 90 min 10 µM nocodazole treatment undergoing FRAP. **k** Quantification of normalized fluorescence recovery. Measurements were taken every 30 s over 10 min ($n = 15$ dendrites RFP-APC2, $n = 12$ dendrites RFP-APC2 + GFP-SpvB, $n = 13$ dendrites RFP-APC2 + Nocodazole, $N = 3$). Graphs represent mean ± SEM. * $p \leq 0.05$. Scale bars represent 10 µm in **c** and **e** zoom out and 5 µm in rest

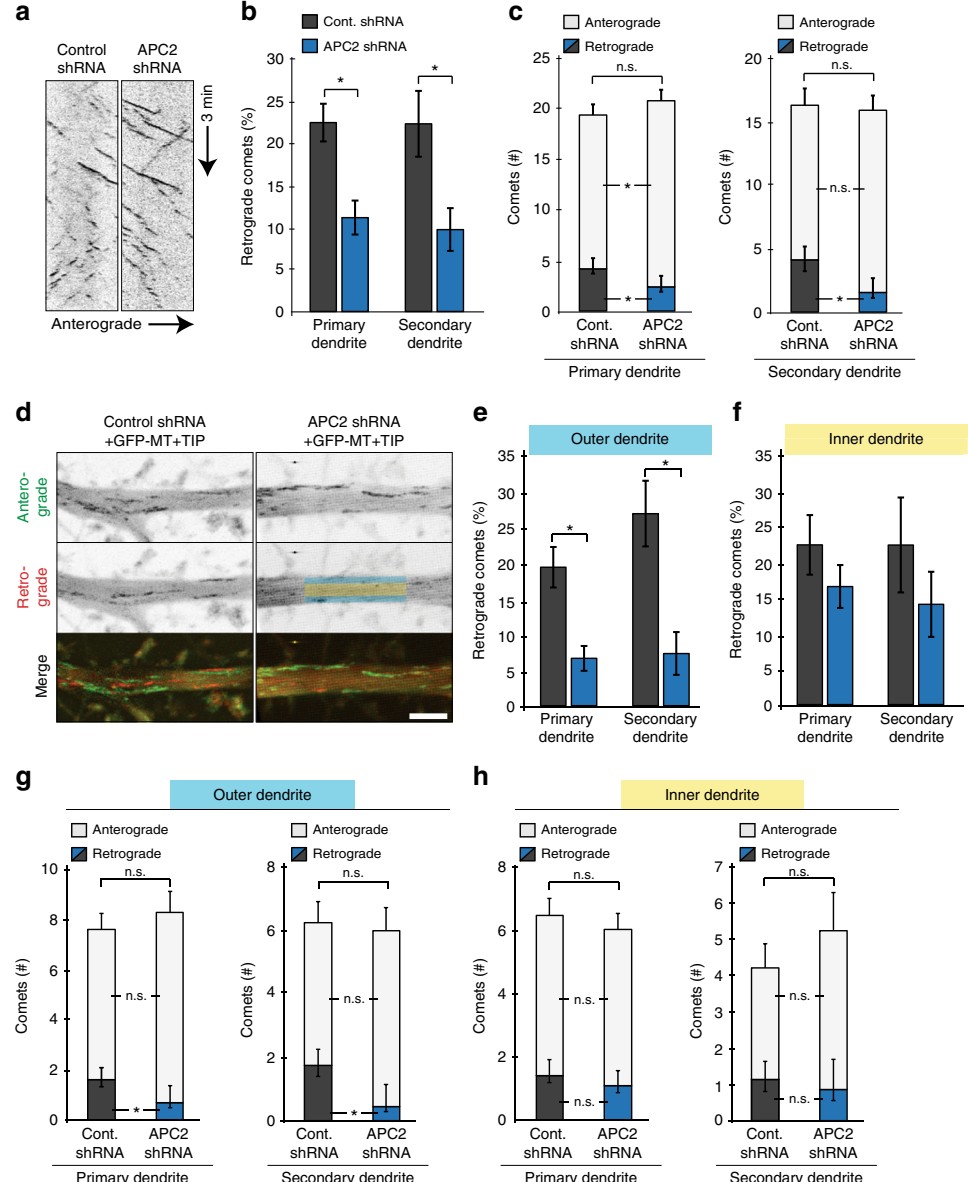

**Fig. 3** APC2 depletion alters minus-end-out microtubule dynamics in dendrites. **a** Representative kymographs of GFP-MT + TIP in neurons co-transfected with either control or APC2 shRNA and imaged at DIV15–16. **b** Percentage retrograde GFP-MT + TIP comets in dendrites in 6.7 μm long regions at least 30 μm away from the cell body in primary dendrite condition and at least 30 μm away from the first branch point in secondary dendrite condition. Calculated from 3 min movies with frames acquired every 0.5 s (n = 41 primary dendrites control shRNA, n = 42 primary dendrites APC2 shRNA, p ≤ 0.001, n = 31 secondary dendrites control shRNA, n = 29 secondary dendrites APC2 shRNA, p = 0.007, N = 3, Mann–Whitney U test). **c** Number of antero- and retrograde GFP-MT + TIP comets (n = 41 primary dendrites control shRNA, n = 42 primary dendrites APC2 shRNA, anterograde p = 0.039, retrograde p = 0.009, n = 31 secondary dendrites control shRNA, n = 29 secondary dendrites APC2 shRNA, anterograde p = 0.159, retrograde p = 0.013, N = 3, Independent samples t-test). **d** Maximum projections of 50 frames of GFP-MT + TIP in control or APC2 depletion conditions color-coded for direction. Green are + TIPs of plus-end-out microtubules, red are + TIPs of minus-end-out microtubules. **e** Percentage retrograde GFP-MT + TIP comets in dendrites in outer dendrite. Color code is from panel d. (n = 31 primary dendrites, p = 0.001, n = 16 secondary dendrites, p = 0.002, N = 3, Mann–Whitney U test). **f** Percentage retrograde GFP-MT + TIP comets in dendrites in inner dendrite (n = 31 primary dendrites control shRNA, n = 30 primary dendrites APC2 shRNA, p = 0.418, n = 14 secondary dendrites control shRNA, n = 15 secondary dendrites APC2 shRNA, p = 0.451, N = 3, Mann–Whitney U test). **g** Number of antero- and retrograde GFP-MT + TIP comets in outer dendrite (n = 31 primary dendrites control shRNA, n = 30 primary dendrites APC2 shRNA, anterograde p = 0.210, retrograde p = 0.003, n = 16 secondary dendrites control shRNA, n = 18 secondary dendrites APC2 shRNA, anterograde p = 0.330, retrograde p = 0.003, N = 3, Mann–Whitney U test). **h** Number of antero- and retrograde GFP-MT + TIP comets in inner dendrite (n = 31 primary dendrites control shRNA, n = 30 primary dendrites APC2 shRNA, anterograde p = 0.833, retrograde p = 0.324, n = 14 secondary dendrites control shRNA, n = 15 secondary dendrites APC2 shRNA, anterograde p = 0.400, retrograde p = 0.591, N = 3, Mann–Whitney U test). Graphs represent mean ± SEM. *p ≤ 0.05. Scale bar is 5 μm in **d**

microtubule network within mature dendrites, perhaps through APC2 clusters inter-binding adjacent microtubules.

Because of increased reorganization of the dendritic cytoskeleton in APC2-depleted neurons, we examined microtubule orientation ratios of all microtubules in the dendrite. Classically, imaging of +TIP markers reveals that about one-third of dynamic microtubules are minus-end-out and two-thirds are plus-end-out[2,5,16,17,29]. To examine orientation of all microtubules we combined live-cell imaging with photo-ablation. This allows for a direct readout of the orientations of microtubules regardless of length or plus-end dynamicity (Fig. 4e)[5,18]. We photo-ablated microtubules in control or APC2-depleted neurons expressing +TIP marker in proximal primary dendrites at DIV15–16 (Fig. 4f). Quantification of percentage plus to minus-end-out microtubules within 6.7 μm in a 3 min movie directly post photo-ablation showed no difference in microtubule orientation ratios between control and APC2 depleted neurons (Fig. 4g).

To further examine the effects of APC2 depletion on dendritic cytoskeleton organization we employed a recently-established single molecule localization technique. By means of super-resolution of dendritic microtubule cytoskeleton we were able to distinguish potential additional effects of treatment, such as changes in bundling between microtubules and reorganization of minus and plus-end-out oriented microtubule bundles. For this we employed motor-PAINT[41]. Neurons were transfected with RFP-tubulin and either control or APC2 shRNA at DIV11 and fixed according to published methods at DIV15[41]. The extracted cytoskeleton of RFP-tubulin positive neurons was then imaged during addition of purified and fluorescently labeled kinesin-1 motors. Kinesin-1 is a well characterized plus-end directed motor, addition of which results in numerous transient runs on individual microtubules[41–43]. We analyzed those running events using single-molecule localization and tracking techniques to construct images of dendritic microtubule arrays in which the orientation of each microtubule is known (Fig. 4h). We found that APC2 depletion did not result in significant differences in microtubule bundle orientations (Fig. 4h, i). These results indicate that APC2 depletion does not lead to complete depolymerization or transportation of specific subsets of microtubules out of the dendrite. Instead, these results support the role of APC2 in balancing dynamics of existing microtubules. Our results indicate that depletion of APC2 affects about half of dynamic portions of minus-end-out microtubule lattices, reducing dynamicity of those microtubules, which we determined is a small proportion of total polymer content, yielding negligible differences in global organization of microtubule polarity.

**Functional domains of APC2's cytoskeleton-interacting region**. Tumor suppressor APC is among the best characterized plus-end-tracking proteins;[44–51] its C-terminus binds to both microtubules and other+TIPs. Because of low sequence homology between APC and APC2 C-termini and their different localizations within cells, APC2 was first thought not to be able to bind microtubules or track plus ends[52]. Currently, it is known that region S within the APC2 C-terminus is necessary and sufficient for microtubule distribution when fused to the rest of the RFP-APC2-ΔC[25]. Moreover, APC2 was found, but not experimentally confirmed, to contain a predicted+TIP binding sequence in the C-terminus[48]. In order to determine the mechanism of physiological RFP-APC2 cluster formation and localization along microtubule bundles (Fig. 2) we turned to Cos7 cells and examined functional domains of the C-terminus and their roles in localizing the protein. Focusing on the microtubule-binding function of the C-terminus, we made two fragments (RFP-APC2-

C1 and RFP-APC2-C2) and subsequently generated several smaller truncations of RFP-APC2-C1 (Fig. 5a). We assessed efficiency of microtubule lattice binding by truncations of RFP-APC2-C1 compared to RFP-APC2-C by measuring the area of truncated protein bound to microtubules compared to total area of the cell harboring microtubules (Fig. 5b).

As previously reported, we found that RFP-APC2 is capable of uniformly decorating microtubules in dividing cells when severely overexpressed (Supplementary Fig. 2g). However, Cos7 cells expressing moderate levels of protein, localize RFP-APC2 in the same manner as neurons (Fig. 2)—in a cluster pattern along microtubules (Fig. 5c, Supplementary Fig. 3a, b). Notably, a small number of clusters localized to actin patches with no microtubule colocalization. RFP-APC2-ΔC localized in aggregates throughout the cell as has been previously described[25]. RFP-APC2-C most efficiently bundled a large proportion of microtubules in the cell. Remarkably, RFP-APC2-C2 is a +TIP efficiently tracking assembling plus ends of microtubules and colocalizing with EB3 (Fig. 5c, Supplementary Fig. 3a, c). We found that smaller truncations of RFP-APC2-C1 lacking the S domain did not colocalize with microtubules, but that RFP-APC2-C4 colocalized with actin stress fibers in HeLa cells (Supplementary Fig. 3d, e). Lastly, we determined that the S domain on its own is not sufficient for binding to microtubules, instead the S domain extended on both sides (RFP-APC2-S2) is the minimum necessary sequence for microtubule interaction (Fig. 5c, Supplementary Figure 3a, d). In summary, our results highlight three cytoskeleton interacting domains within the C-terminus of APC2: an extended S microtubule binding domain, an actin binding domain and a +TIP binding domain. When all three are present together in the APC2 protein, microtubule binding overrules the rest.

**Mechanism of APC2 C-terminus MT binding**. In order to determine the mechanism behind APC2's binding to microtubules, we focused on RFP-APC2-C, RFP-APC2-C1, and RFP-APC2-C2. First, we aimed to determine whether RFP-APC2-C bundles microtubules in a specific direction. We co-expressed RFP-APC2-C and GFP-MT + TIP in Cos7 cells and compared dynamics and orientation of microtubule bundles to known antiparallel bundler PRC1 and parallel bundler TRIM46 (Fig. 6a). We photo-ablated microtubule bundles to check whether the bundles had a preferred microtubule orientation and quantified the amount of time it took for microtubules to repolymerize as a measure of stabilizing strength of the bundler. RFP-APC2-C bundled microtubules were not organized in a preferred aligned direction, showing no directional organization properties of the C-terminus. Interestingly, RFP-APC2-C bundled microtubules took significantly longer to repolymerize than both PRC1 and TRIM46 bundled microtubules (Fig. 6b). The lag was completely abolished when we mutated the + TIP binding motif SxLP to SxNN, suggesting that the function of the + TIP interaction of APC2 is to catch and, perhaps, slow microtubule polymerizing ends (Fig. 6a, b). Next, we found that RFP-APC2-C was enriched on tyrosinated microtubules in Cos7 cells when compared to acetylated microtubules (Fig. 6c, d). And so was RFP-APC2-C1, although to a lesser degree (Fig. 6d). It is important to note that tyrosination and acetylation are not mutually exclusive. Acetylation has long been considered a sign of microtubule longevity. Newly polymerized microtubules are tyrosinated and not acetylated. Long-lived microtubules gradually lose tyrosination and become enriched in acetylation[53]. These results suggest that the C-terminus of APC2 may localize to freshly assembled dynamic microtubules and stabilize them upon binding, which in turn increases their acetylation level with time (Supplementary Fig. 3f).

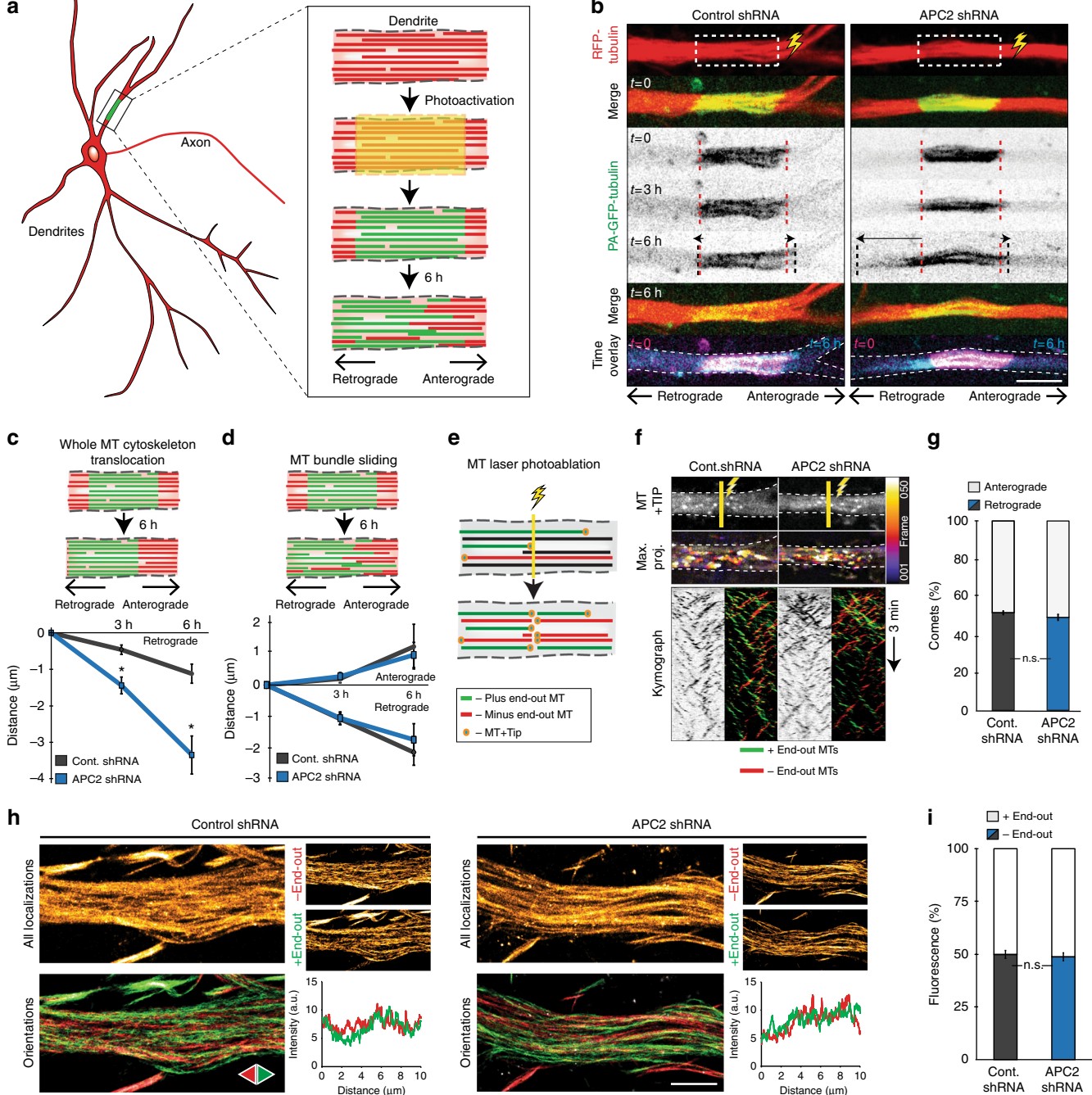

**Fig. 4** APC2 effect on global organization of microtubule polarity in dendrites. **a** Neurons transfected with RFP-tubulin, PA-GFP-tubulin and either control or APC2 shRNA on DIV11–12 were live imaged on DIV15–16 (schematic drawn by O.I.K.). **b** Representative images of control or APC2 depleted neuron with photoactivated region of dendrite undergoing microtubule bundle sliding. Red dotted line represents edges of photoactivated region at $t = 0$. Black dotted line represents edges photoactivated region at the corresponding time point. Time overlay has frame at $t = 0$ in magenta merged with $t = 6$ h in cyan. **c** Translocation distance of whole photoactivated region towards the cell body was measured ($n = 21$ dendrites control shRNA, $n = 26$ dendrites APC2 shRNA, $N = 3$, Two-Way ANOVA, Tukey post-hoc). **d** Distance of microtubule bundles sliding out of the photoactivated region both towards and away from cell body was measured ($n = 8$ dendrites control shRNA, $n = 14$ dendrites APC2 shRNA, $N = 3$, Two-Way ANOVA, Tukey post-hoc). **e** Schematic representation of microtubule photo-ablation in dendrites. **f** Neurons were transfected with GFP-MT + TIP and either control or APC2 shRNA at DIV11–12 and imaged DIV15–16. Proximal dendrites were photo-ablated 30 μm away from cell body and microtubule orientations were imaged for 3 min. Representative images of GFP-MT + TIP still frames, color-coded maximum projections and kymographs coded for direction of comets are shown. **g** Percentage + TIP comets in dendrites toward cell body post photo-ablation. ($n = 14$ dendrites control shRNA, $n = 17$ dendrites APC2 shRNA, $p = 0.199$, $N = 2$, Independent samples $t$-test). **h** Microtubule motor based super-resolution reconstruction of segments of proximal dendrites of neurons at DIV15 transfected with RFP-tubulin and either control or APC2 shRNA. Signal shown is derived from motor-PAINT. Red microtubules are minus-end-out, green are plus-end-out. Line scans are of green and red signals corresponding to microtubule orientations along 10 μm segments of dendrites. **i** Percentage mean red intensity to total was quantified as a measure of percentage minus-end-distal microtubules in the control or APC2 depleted neurons ($n = 5$ dendrites control shRNA, $n = 7$ dendrites APC2 shRNA, $p = 0.588$, $N = 2$, Independent samples $t$-test). Graphs represent mean ± SEM. *$p \leq 0.05$. Scale bars represent 5 μm in **b** and 2 μm in **h**

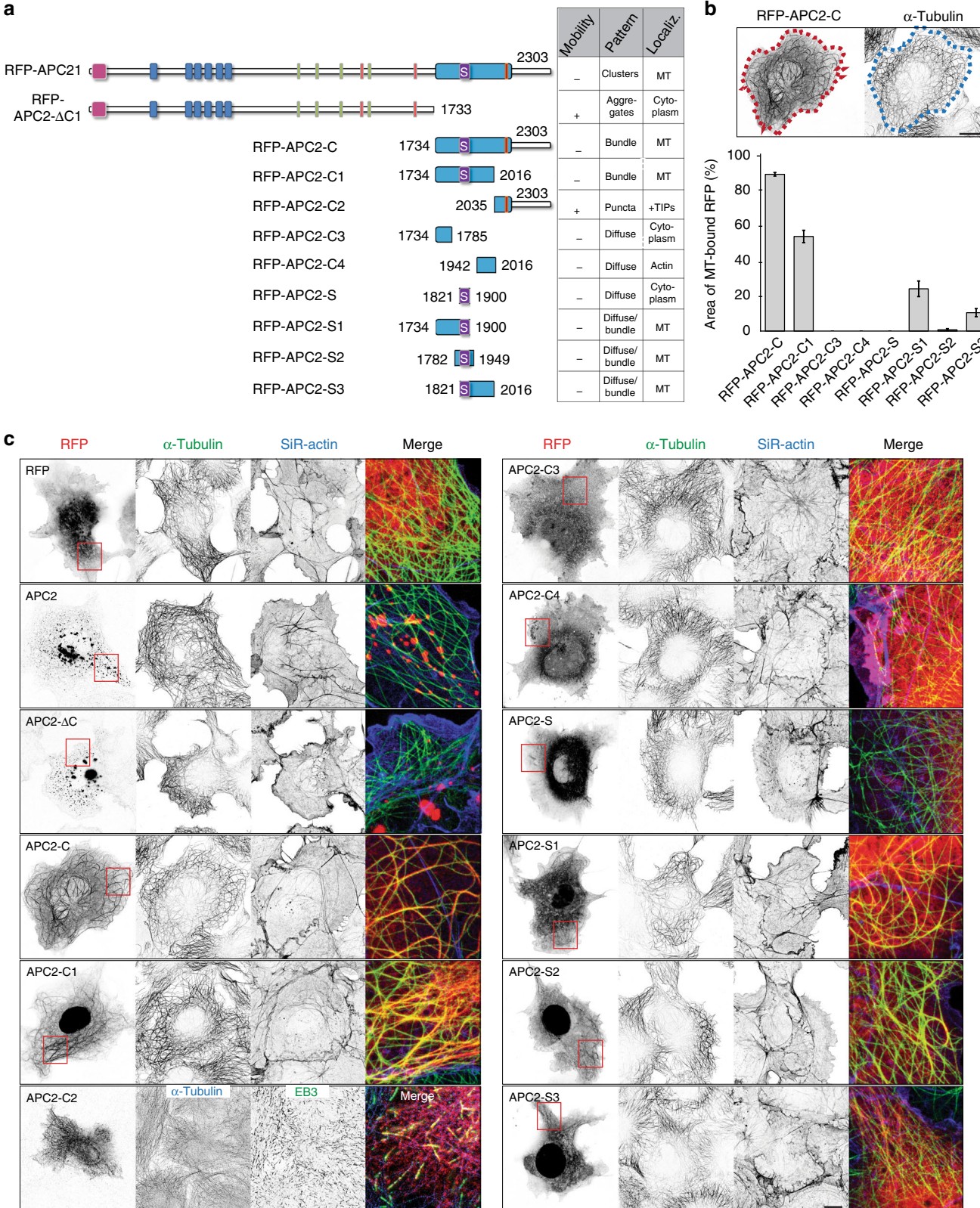

**Fig. 5** Functional domains of APC2 cytoskeleton interacting region. **a** Schematic representation of human APC2 protein truncations generated for these experiments. Table shows whether the expressed truncations are mobile, what pattern their localization follows and what region of cell they are localized to. **b** Representative images of area of microtubules decorated with RFP-APC2-C in Cos7 cells (red dotted line) and total area occupied by microtubules of same cell (blue dotted line) are shown. Percentage of area of microtubule bound construct (e.g., red dotted line) over total microtubule occupied area of cells (e.g., blue dotted line) was measured and used to determine affinity of the truncations for microtubules ($n = 50$ cells, $N = 2$). **c** Representative images of truncated proteins expressed in Cos7 cells for 24 h before fixation and staining for tubulin, actin and EB3. Graphs represent mean ± SEM. Scale bars are 10 μm

Previously, region S was shown to target full-length APC2 to microtubules[25]. Here, we wanted to analyze the precise molecular mechanism by which the whole C-terminus binds to micro-tubules. For this we designed an inducible FKBP-rapalog-FRB heterodimerization assay in Cos7 cells to test APC2's precise role in microtubule binding[54]. We expressed a membrane bound (harboring a CAAX motif) FKBP domain together with FRB bound to either RFP-APC2-C, RFP-APC2-C1 or RFP-APC2-C2.

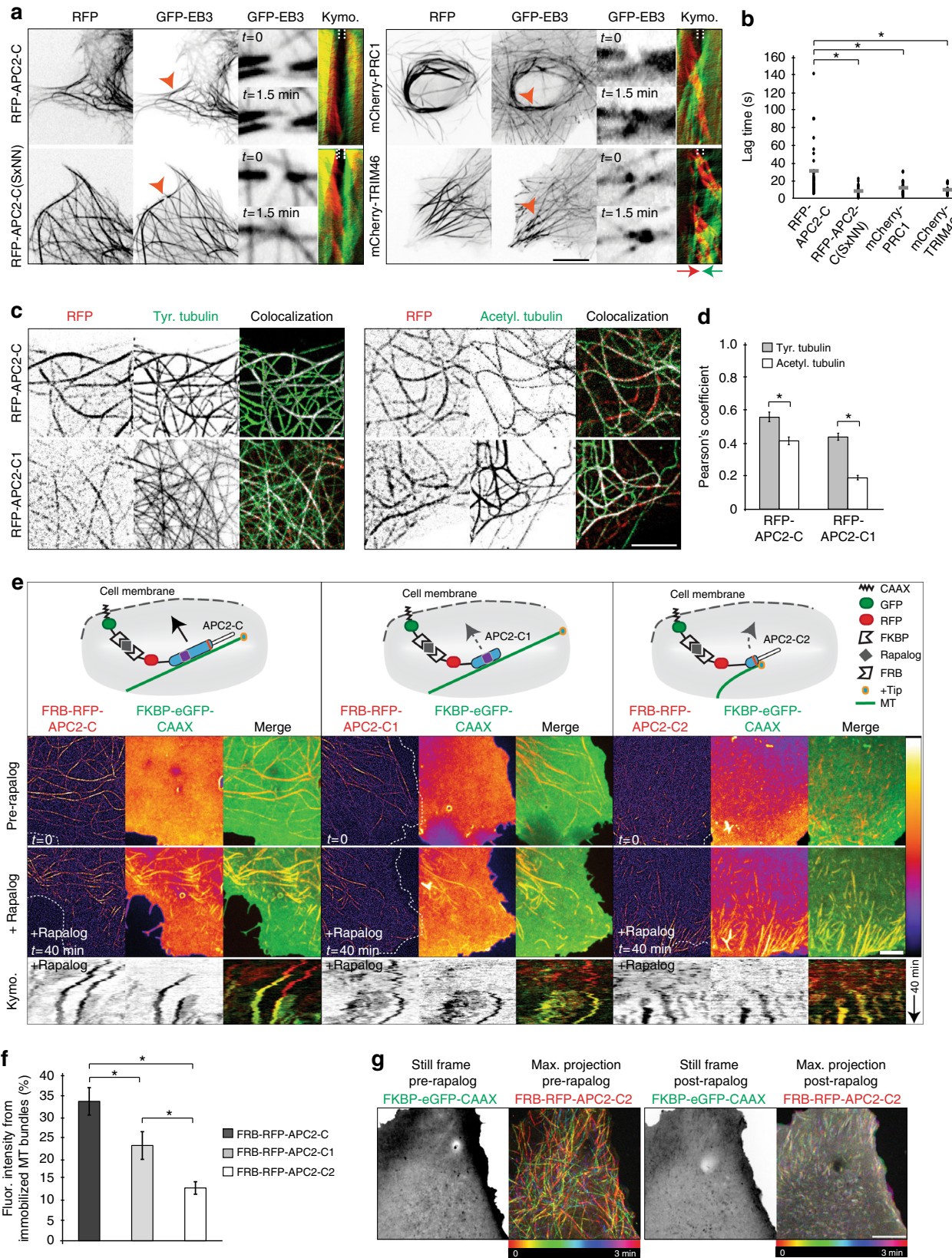

During live imaging, we added rapalog to heterodimerize FKBP-FRB (Fig. 6e). Upon rapalog addition, FRB-RFP-APC2-C underwent the least amount of diffusion and recruitment to the membrane, and instead remained enriched on microtubules. Within 40 min of treatment with rapalog, FKBP-eGFP-CAAX showed the greatest membrane displacement due to long stretches of FRB-RFP-APC2-C-decorated microtubule bundles being recruited by CAAX (Fig. 6f, Supplementary Movie 2). FRB-RFP-APC2-C1 underwent more diffusion off of microtubule bundles, compared to FRB-RFP-APC2-C. Furthermore, the 40 min treatment resulted in fewer and smaller stretches of FRB-RFP-APC2-C1-decorated microtubule bundles being recruited by CAAX. The majority of bundles recruited by FRB-RFP-APC2-C and FRB-RFP-APC2-C1 were acetylated, which suggests that both constructs are able to stabilize microtubule bundles and strongly bind to acetylated microtubule lattice (Supplementary Fig. 4a, b)[20]. Curiously, FRB-RFP-APC2-C2 behaved very differently upon addition of rapalog. Pre-rapalog FRB-RFP-APC2-C2 tracked plus ends and within 40 min of rapalog addition became diffuse and evenly heterodimerized to FKBP-eGFP-CAAX. However, as dynamic microtubules encountered the cell membrane, FRB-RFP-APC2-C2 caught and bound + TIPs that were tracking polymerizing ends of microtubules via the SxLP motif and immobilized stretches of newly assembled microtubules on the membrane surface (Fig. 6e–g, Supplementary Movie 2). These stretches were composed of tyrosinated microtubules, positive for EB3 and remained immobile for the duration of imaging (3 min live imaging post 40 min rapalog treatment) (Fig. 6g, Supplementary Fig. 4c). These results demonstrate that the C-terminus of APC2 binds microtubules most efficiently in the presence of the microtubule binding domain and the + TIP tracking domain. Furthermore, FRB-RFP-APC2-C2 harboring the + TIP tracking domain can capture and immobilize freshly polymerized microtubule plus ends when heterodimerized to the membrane or potentially as part of the full length APC2 protein.

**APC2 clustering is driven by DYNLL2-binding.** In dendrites, RFP-APC2 is localized in a regular cluster pattern along microtubule bundles (Fig. 2). We determined that the proper cluster formation and localization to microtubules is driven by the C-terminus, since RFP-APC2-ΔC loses the localization and size restriction of the clusters. We then screened the C-terminus for characterized binding motifs and found two potential interacting sites with the LC8 complex (composed of DYNLL1 or DYNLL2)[55], one within region S [SSSSQT] and one in C2 [DAVVQT]. The mammalian light chains DYNLL1 and DYNLL2 form homodimers and share 93% identity, differing by only six

amino acids[56]. Apart from its function in the dynein motor complex, LC8 also associates with a large variety of proteins to serve as a dimerizing hub for a variety of complexes[57]. To determine if the LC8 complex interacts with APC2, we co-expressed RFP-APC2 with GFP, GFP-DYNLL1, GFP-DYNLL2 or GFP-DYNC1LI2. DYNC1LI2 is a light intermediate chain, which also forms homo-oligomers and binds to the N-terminal base of the dynein heavy chain with a primary role for cargo binding[58]. We found that RFP-APC2 clusters in neurons colocalize with GFP-DYNLL2 and to a significantly lesser degree with GFP-DYNLL1 (Fig. 7a, Supplementary Fig. 4d). There was no colocalization with GFP or GFP-DYNC1LI2, suggesting that RFP-APC2 is not acting as a cargo to dynein, but as a substrate specifically for LC8. Furthermore, we found that GFP-DYNLL2 is diffuse when co-expressed with RFP, which suggests that it does not play a role in recruitment of RFP-APC2, but instead that RFP-APC2 recruits GFP-DYNLL2 (Fig. 7a). To determine the binding site of RFP-APC2-C to LC8 we performed pull-downs on truncated C-terminus of APC2 (Fig. 7b, c). Both RFP-APC2-C1 and RFP-APC2-C2, containing potential interacting sites, were pulled down by BioGFP-DYNLL2 and to a lesser degree by BioGFP-DYNLL1. To further exclude potential for interaction of APC2 with cytoplasmic dynein motor, we performed a pull-down with the heavy chain of dynein (BioGFP-DYNC1H1) and found that RFP-APC2-C does not interact with it in the same manner as adapter protein dynactin (p150) (Fig. 7d).

To determine the importance of the two interaction sites in binding to DYNLL2, we mutated them within the C-terminus, yielding RFP-APC2-C(c1SQT-AAA) and RFP-APC2-C(c2VQT-AAA). Interestingly, in Cos7 cells RFP-APC2 and RFP-APC2-C recruited GFP-DYNLL2 to microtubules and so did RFP-APC2-C(c1AAA), but not RFP-APC2-C(c2AAA) (Fig. 7e, f). Indeed, BioGFP-DYNLL2 could pull down RFP-APC2-C(c1AAA), but not RFP-APC2-C(c2AAA), determining that the second binding site is vital for interaction with DYNLL2 (Fig. 7g). To define importance of DYNLL2 binding to APC2 in neurons for its cytoskeleton-related function, we mutated the C2 site in full length APC2 and expressed it in neurons (Fig. 7h). RFP-APC2 (c2AAA) localized in dendrites in small aggregates and lacked regular organization as seen with RFP-APC2 (Figs. 7h and 2d). We also checked for localization defects of RFP-APC2 in DYNLL2 depleted neurons and found that RFP-APC2 clusters are no longer regularly spaced when compared to control (Supplementary Fig. 4e), suggesting that DYNLL2 binding is vital for proper APC2 function in dendrites. Finally, in order to determine if APC2 cytoskeleton interaction and proper cluster assembly via DYNLL2 are important for neuronal dendritic development we performed a morphology rescue experiment and

**Fig. 6** Mechanism of APC2 C-terminus microtubule binding. **a** Cos7 cells were transfected with GFP-EB3 and either RFP-APC2-C, RFP-APC2-C(SxNN), mCherry-PRC1 or mCherry-TRIM46. One microtubule bundle per cell was photo-ablated and imaged for 90 s stream for microtubule polymerization. Kymographs are color-coded for opposite directions of assembling microtubule plus ends. **b** Lag time from photo-ablation to first spotted microtubule assembly via GFP-EB3 was measured from live movies ($n = 52$ bundles RFP-APC2-C, $n = 19$ bundles RFP-APC2-C(SxNN), $n = 28$ bundles mCherry-PRC1, $n = 23$ bundles mCherry-TRIM46, $N = 2$, Kruskal–Wallis Test, Dunn's post-hoc). **c** Cos7 cells expressing either RFP-APC2-C or RFP-APC2-C1 were fixed and immunostained for tyrosinated or acetylated tubulin. Colocalized pixels are white, while non-colocalized RFP pixels are red and non-colocalized tubulin pixels are green. **d** Pearson's coefficient was quantified from colocalization of RFP and GFP channel from maximum of three different regions per cell ($n = 24$ areas RFP-APC2-C tyr. tubulin, n = 28 areas RFP-APC2-C acetyl. tubulin, n = 30 all other conditions; tyr. tubulin $p < 0.001$, $N = 3$, Independent samples $t$-test, acetyl. tubulin $p < 0.001$, Independent samples $t$-test). **e** Cos7 cells were transfected with FKBP-eGFP-CAAX and either FRB-RFP-APC2-C, FRB-RFP-APC2-C1 or FRB-RFP-APC2-C2 and imaged 24 h later. Kymographs are of 40 min movies with frames taken every 30 s post rapalog addition. RFP and GFP channels are presented in fire scale (schematic drawn by O.I.K.). **f** Percentage of fluorescence arising from bundles localized to FKBP-eGFP-CAAX at the cell membrane 40 min post rapalog addition was measured by subtracting mean fluorescence from non-bundled regions of FKBP-eGFP-CAAX from total mean fluorescence intensity of the cell ($n = 18$ cells FRB-RFP-APC2-C and C2, $n = 16$ cells FRB-RFP-APC2-C1, N = 2, One-Way ANOVA, Tukey post-hoc). **g** Representative still frames and maximum projections of 3 min movies taken every 0.5 sec from the same cell expressing FKBP-eGFP-CAAX and FRB-RFP-APC2-C2 before and after rapalog addition. Graphs represent mean ± SEM. *$p \leq 0.05$. Scale bar is 10 μm in **a** and 5 μm in **c**, **e** and **g**

a dynamic microtubule orientation rescue experiment. We found that neither APC2 lacking the C-terminus, nor APC2 with mutated DYNLL2 binding site c2 were able to rescue APC2 depletion driven defects in dendritic arborization (Fig. 7i, Supplementary Fig. 2a). These results demonstrate that

interaction of APC2 with the dendritic cytoskeleton and its proper cluster organization and localization are necessary for proper dendrite development, and that these properties depend on cytoskeleton-binding domains within the C-terminus and interaction with DYNLL2.

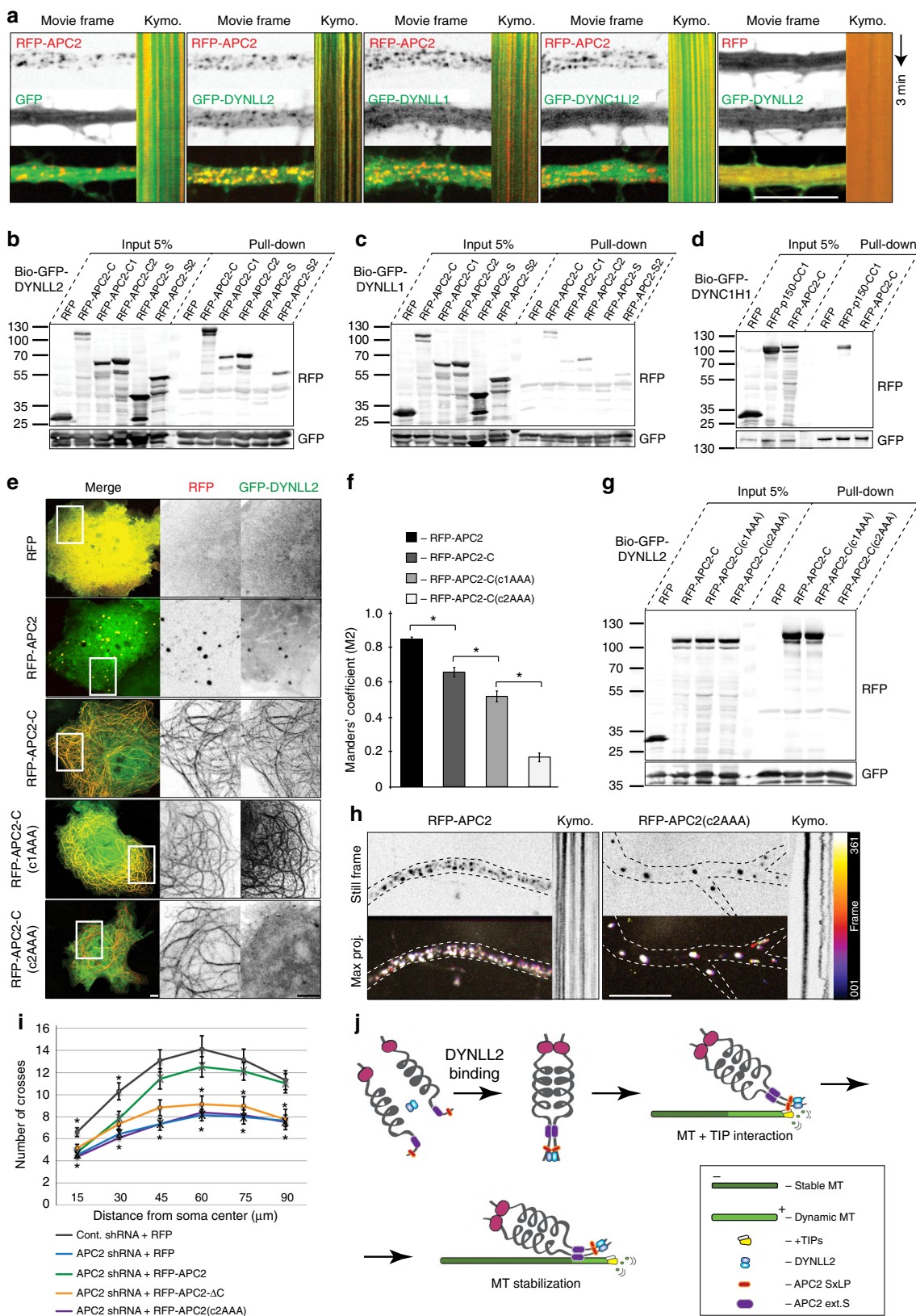

## Discussion

Here, we investigated the role of APC2 in regulation of vertebrate dendrite development. Depletion of APC2 in neurons results in significant morphological changes in both the dendritic tree and the dendritic spines. We found that APC2 depletion markedly affects dendritic microtubule dynamics, which are essential for dendritic outgrowth and dendritic spine dynamics. Polymerizing plus ends of microtubules can enter spines in an activity-dependent manner and act as a direct route for microtubule-based motor-driven transport of synaptic cargo into spines[30,59]. Intriguingly, APC2 depletion severely affects a subset of microtubules. We determined those microtubules to be dynamic minus-end-out microtubules localized primarily to the outer edge of the dendritic shaft. Typically, in vertebrate dendrites microtubules are equally mixed in orientation. Recent work has shown that the microtubule arrays are spatially segregated both by orientation and post-translational modifications. Minus-end-out microtubules are more stable and are preferentially localized to the inner dendrite, while a larger proportion of the plus-end-out microtubules is dynamic and preferentially localized to the outer dendrite[5,29,41]. Our results suggest that APC2 promotes dynamics of minus-end-out microtubules localized to the outer dendrite. We speculate that APC2 might recognize innate differences in structure, length, dynamics and/or longevity between plus and minus-end-out microtubules in dendrites. Our data supports a model in which APC2 regulates dendrite development by clustering and anchoring along longer freshly polymerized ends of microtubules, where it may act as a potential rescue point, thereby promoting dynamicity of those microtubules.

We found that the mechanism of strikingly regular and clustered localization of APC2 depends on its C-terminus. C-terminus of APC2 has been previously shown to affect protein function and localization, since a frameshift mutation leads to loss of function of the protein in developmental disorder Sotos syndrome. Sotos syndrome patients present with multiple neurological symptoms, including Autism Spectrum Disorder, which have in part been attributed to APC's role in neuronal development and function[25,60]. When we deleted the C-terminus of APC2, the truncated protein aberrantly dispersed throughout the dendrites in large aggregates that often underwent transport-like movements. We found that truncated C-terminus efficiently bundles microtubules and is characterized by three cytoskeleton interacting domains. The SxLP motif interacts with growing microtubule tips, while extended S domain binds the microtubule lattice. Together, the two domains create an efficient microtubule binding C-terminus, which acts as a bundler and no longer tracks + TIPs. The third domain localizes to actin stress fibers; however, this association is overruled in the presence of microtubule binding. We also determined that previously characterized S domain cannot bind microtubules on its own, but requires additional flanking sequence on both sides. Consequently, we identify the microtubule binding domain as extended S. Our membrane recruitment assay suggests a mechanism of how APC2 C-terminus binds to microtubules. Conceivably, as APC2 encounters a polymerizing microtubule, SxLP motif interacts with + TIPs and brings extended S microtubule-binding domain in contact with the lattice. After a short delay, microtubule polymerization rate overpowers interaction with SxLP motif and the microtubule continues to polymerize further, while APC2 remains stably bound to the freshly polymerized microtubule through the extended S domain. It is plausible, that once that microtubule undergoes catastrophe, the stabilizing patch of APC2 allows for microtubule regrowth from that point on, ensuring dynamics of the microtubule.

Remarkably, proper clustering and localization of APC2 in dendrites is dependent on its interaction with DYNLL2. DYNLL2 is LC8-type 2 cytoplasmic dynein light chain, which is a well-characterized dimerization and structuring hub[57,61]. LC8 is most well-known for its function in the cytoplasmic dynein motor, but it also associates with a large variety of proteins, which occupy the same binding grooves as the intermediate chain. For example, LC8 has been shown to promote self-association of Swallow protein to a stable dimer, which promotes its conformational stability and orders the structure at the C-terminus domain necessary for Swallow localization and transport along the cytoskeleton[57]. We show that DYNLL2 binds a critical sequence in RFP-APC2-C2 and also interacts with a non-critical sequence in RFP-APC2-C1. It is conceivable that DYNLL2 in a manner zips the C-terminus tail, starting from the C2 and continuing to the C1 site, thus structuring and stabilizing the normally unstructured tail and opening the microtubule interacting domains for contact with the cytoskeleton (Fig. 7j). Mechanistically, our data suggests that in vertebrate dendrites DYNLL2-dependent structuring of APC2's C-terminus drives APC2 to localize to long dynamic portions of microtubules through its interaction with + TIPs that leads to subsequent stabilization of the microtubule lattice via the extended S domain.

Future work is needed to explore the effect of APC2 clustered localization in dendrites on its downstream effectors and perhaps on local actin regulation, as well as, regulation of APC2 by phosphorylation or interaction with partner proteins[62–64]. Finally, investigating the potential link between APC2's minus-end-out microtubule dynamics-promoting function and neuronal activity would provide insight into the role of dynamic minus-end-out microtubules in activity-dependent dendrite remodeling.

## Methods

**Animals**. All experiments were approved by the DEC (Dutch Experimental review Committee), performed in line with institutional guidelines of the University

---

**Fig. 7** APC2 clustering is driven by DYNLL2 binding. **a** Neurons were transfected with RFP-APC2 and either GFP, GFP-DYNLL2, GFP-DYNLL1 or GFP-DYNC1LI2 or RFP and GFP-DYNLL2 and imaged 24 h later. **b** Streptavidin pull-down assay with BioGFP-DYNLL2 and the indicated APC2-C truncations, expressed in HEK293T cell lysates, analyzed by Western blotting with the indicated antibodies. **c** Streptavidin pull-down assay with BioGFP-DYNLL1 and the indicated APC2-C truncations, expressed in HEK293T cell lysates, analyzed by Western blotting with the indicated antibodies. **d** Streptavidin pull-down assay with BioGFP-DYNC1H1 and the indicated APC2-C truncations, expressed in HEK293T cell lysates, analyzed by Western blotting with the indicated antibodies. **e** Cos7 cells were transfected with GFP-DYNLL2 and either RFP, RFP-APC2, RFP-APC2-C, RFP-ACP2-C(c1AAA) mutation or RFP-ACP2-C (c2AAA) mutation, fixed 24 h later and imaged. GFP-DYNLL2 co-transfected with RFP-APC2 and imaged live after 24 h. **f** Thresholded Manders' coefficient was measured from max two areas per cell in 10 × 10 μm ROIs. ($n = 25$ areas for RFP-APC2, $n = 32$ areas for RFP-APC2-C, $n = 30$ areas for RFP-APC2-C(c1AAA), $n = 35$ areas for RFP-APC2-C(c2AAA), $N = 2$, One-Way ANOVA, Tukey post-hoc). **g** Streptavidin pull-down assay with BioGFP-DYNLL2 and the indicated C-terminus mutations, expressed in HEK293T cell lysates, analyzed by Western blotting with the indicated antibodies. **h** Neurons were transfected with either RFP-APC2 or RFP-APC2(c2AAA) and dendrites were imaged 24 h later at DIV15–16. **i** Sholl analysis of dendritic branching at DIV11 ($n = 34$ neurons control shRNA + RFP, $n = 35$ neurons APC2 shRNA + RFP, $n = 28$ neurons APC2 shRNA + RFP-APC2, $n = 34$ neurons APC2 shRNA + RFP-APC2-ΔC, $n = 20$ neurons APC2 shRNA + RFP-APC2(c2AAA), $N = 3$, Two-Way ANOVA, Tukey post-hoc). **j** Model for APC2 interaction with microtubules (schematic drawn by O.I.K.). Scale bar is 10 μm in **a** and 5 μm in **e** and **h**

Utrecht and were conducted in agreement with Dutch law (Wet op de Dierproe-ven, 1996) and European regulations (Guideline 86/609/EEC). Female pregnant Wistar rats (Janvier) were aged at least 10 weeks at the time of delivery. Upon delivery, they were kept at SPF facilities in a controlled 12 h light-dark cycle with a temperature of $22 \pm 1$ °C and were given unrestricted access to food and water. The animals were housed in small groups in transparent plexiglass cages with wood-chip bedding and paper tissue for nest building.

**Primary neuronal cultures.** Primary hippocampal cultures were prepared from embryonic day 18 rat brains (both genders)[30,65]. None of the parameters analyzed in this study are reported to be affected by embryo gender. Cells were plated on coverslips coated with poly-L-lysine (37.5 µg/mL) and laminin (1.25 µg/mL) at a density of 100,000/coverslip. Neurons were cultured in Neurobasal medium (NB) supplemented with 2% B27 (GIBCO), 0.5 mM glutamine (GIBCO), 15.6 µM glu-tamate (Sigma), and 1% penicillin/streptomycin (GIBCO) at 37 °C in 5% $CO_2$.

**Cell culture.** African Green Monkey SV40-transformed kidney fibroblast (Cos7) and human epithelial cervical adenocarcinoma cells (HeLa) were obtained from ATCC and cultured in DMEM supplemented with 10% FCS and 1% penicillin/streptomycin on 18 mm glass coverslips at 37 °C and 5% $CO_2$. Human embryonic kidney cells 293 (HEK) were from ATCC and cultured in DMEM/Ham's F10 (50:50) supplemented with 10% FCS and 1% penicillin/streptomycin at 37 °C and 5% $CO_2$. Cells were plated in 10 cm dishes. Cell lines were not authenticated by authors after purchase. All cell lines routinely tested negative for mycoplasma.

Cos7 cells were used for localization studies, because they tend to be quite flat and cytoskeleton proteins can be easily imaged. HeLa cells were used for actin stretch fiber localizations. HEK cells were used for pull-down experiments due to their high levels of expressed protein production.

**DNA and shRNA constructs.** The following vectors and constructs have pre-viously been described and kindly provided: pSuper vector[66], Bio-GFP expression vector[30], mCherry-TRIM46 and mCherry-PRC1[18], MARCKS-BFP[67], GFP-SpvB[36], PA-GFP-α-tubulin[68], GFP-EB3[29], mRFP-p150-CC1[68], protein-biotin ligase BirA[69], pSuper.puro EV (Dr. R. Poot, Erasmus University), DmKHC(1–421)-GFP-6xHis[41]. Human APC2 clone was kindly provided by Dr. Eva Wenzel (Oslo University Hospital). All APC2 expression constructs were generated by a PCR-based strategy and placed in βactin vector[70]. TagRFP-T was placed at the N-terminus of all APC2 expression constructs (here referred to as RFP). C-terminus truncations were based on Almuriekhi et al.[25]. Mutation constructs were mutated as follows using a PCR based strategy: RFP-APC2-C(SxNN) from [SPSRLPV] to [SPSRNNV]; RFP-APC2-C(c1AAA) from [SSSSQT] to [SSSAAA]; and RFP-APC2-C(c2AAA) from [DAVVQT] to [DAVAAA]. APC2 shRNA (5′-CCTGATCTACAGTGTACAC-3′) and DYNLL2 shRNA (5′-GTTGCAATCCTCCTCTTCA-3′) were inserted in pSuper vectors. Empty pSuper vector was used as control. For qPCR, puromycin selection cassette was taken from pSuper.puro EV and inserted into the APC2shRNA pSuper vector. GFP-MT + TIP construct used for live-imaging stu-dies to track polymerizing ends of microtubules was synGFP-MACF18 (with Synapsin promoter) generated using published strategy to reduce expression levels, since we used it in conjunction with shRNA expressed over 4 days[71]. Briefly, upstream open reading frame containing a Kozak sequence was introduced before the start codon of GFP in the synGFP-MACF18 construct. Depending on strength of the Kozak sequence, a certain percent of the ribosomes are expected to initiate translation here and disassemble once they reach the stop codon, therefore the number of ribosomes that can translate the gene of interest is lowered and expression becomes significantly reduced. We used Kozak sequence [gggATG] followed by [GGTTAA]. TagRFP-α-tubulin was subcloned from pTagBFP-α-tubulin (gift from Evrogen) into βactin vector. FRB-FKBP heterodimerization system was previously described[54]. FRB was inserted via PCR-based strategy in front of TagRFP-T in APC2-C, APC2-C1 and APC2-C2 constructs. FKBP-eGFP-CAAX was cloned in βactin vector. Dynein clones were kindly provided by Dr. Chin-Yin Tai (University of Massachusetts Medical School) and subsequently cloned into BioGFP pEGFP-C2 vectors by Dr. M. Kuijpers (Utrecht University). BioGFP-DYNC1H1 includes rat sequence of dynein heavy chain 1–1140 a.a. The rest of the dynein constructs are based on full length human sequence.

**Antibodies and reagents.** The following antibodies were used in this study: rabbit anti-APC2 (in paper referred to as Ab#1) (1:200, Abcam, ab80018, RRID: AB_2040524), rabbit anti-APC2 (in paper referred to as Ab#2) (1:500, Sigma-Aldrich, SAB3500185, RRID:AB_10638135), mouse anti-MAP2 (1:500, clone HM-2; Sigma-Aldrich, M9942, RRID:AB_477256), chicken anti-MAP2 (1:10,000, Abcam, ab5392, RRID:AB_2138153), rabbit anti-GFP (1:10,000, Abcam, ab290, RRID:AB_303395), rabbit anti-mRFP (1:5,000, Chemicon, AB3216), rabbit anti-TagRFP (1:5,000, Evrogen, AB233, RRID:AB_2571743), rabbit anti-homer1 (1:500, Synaptic Systems, 160 002, RRID:AB_2120990), mouse anti-bassoon (1:500, Enzo Life Sciences, SAP7F407, RRID:AB_2313990), mouse anti-α-tubulin (1:800, Sigma-Aldrich, T-5168, RRID: AB_477579), mouse anti-acetylated tubulin (1:1,000, Sigma-Aldrich, T7451, RRID: AB_609894), rat anti-tubulin-tyrosinated (1:800, YL1/2, Abcam, ab6160, RRID: AB_305328), rabbit anti-TRIM46[18] (1:500), mouse anti-GM130 (1:500, BD Transduction Laboratories, 610823, RRID: AB_398142), mouse

anti-Rab6 (1:250, a gift of A. Barnekow, University of Muenster, Germany), mouse anti-β-catenin (1:250, BD Biosciences, 610153, RRID:AB_397554), rabbit anti-Axin (1:250, C-terminal; Sigma-Aldrich, A0481, RRID:AB_796193), rabbit anti-Camsap-2 (1:250, Proteintech Group, 17880–1-AP, RRID:AB_2068826), mouse anti-Rab11 (1:500, BD Biosciences, 610656, RRID:AB_397983), rabbit anti-LAMTOR4 (1:500, Atlas Antibodies, HPA020998, RRID:AB_1845762), rabbit anti-EB3[29](1:500).

Secondary antibodies were used at 1:500 concentrations, unless specified otherwise, as follows: anti-chicken Alexa 488 (Thermo Fisher Scientific, A11039, RRID:AB_142924), anti-mouse Alexa 488 (Thermo Fisher Scientific, A-11029, RRID:AB_2534088; A-21121, RRID:AB_2535764; A-21131 RRID:AB_2535771; A-21141, RRID:AB_2535781), anti-rabbit Alexa 488 (Thermo Fisher Scientific, A-11034, RRID:AB_2576217), anti-rat Alexa 488 (Thermo Fisher Scientific, A-11006, RRID:AB_2534074), anti-mouse Alexa 647 (Thermo Fisher Scientific, A-21240, RRID:AB_2535809; A-21241, RRID:AB_2535810; A-21242, RRID:AB_2535811), anti-rabbit Alexa 647 (Thermo Fisher Scientific, A-21245, RRID:AB_2535813), anti-rat Alexa 647 (Thermo Fisher Scientific, A-21247, RRID:AB_141778), anti-mouse IRdye680LT (1:20,000, LI-COR Biosciences, 926–68020, RRID: AB_10706161), anti-rabbit IRdye680LT (1:20,000, LI-COR Biosciences, 926–68021, RRID:AB_10706309), anti-mouse IRdye800CW (1:15,000, LI-COR Biosciences, 926–32210, RRID:AB_621842), anti-rabbit IRdye800CW (1:15,000, LI-COR Biosciences, 926–32211, RRID:AB_621843).

Other reagent used was SiR-actin (Spirochrome, CY-SC001).

**Primary neuron transfections.** For live imaging and immunocytochemistry experiments adhered hippocampal neurons were transfected using Lipofectamine 2000 (ThermoFisher, 11668019) at time points specified in each figure legends. Briefly, DNA (0.5 µg for expressing constructs overnight, 0.3 µg for expressing constructs over 4 days and 0.7 µg for shRNA expression) was mixed with 3.3 µL of Lipofectamine 2000 in 200 µL NB (pre-incubated for 30 min) for 10 min, and then added to the neurons in NB at 37 °C in 5% $CO_2$ for 45 min. Next, neurons were washed once with NB and transferred to their original medium at 37 °C in 5% $CO_2$ for 24 h (for protein expression), 96 h (for APC2 shRNA expression) and 48 h (for DYNLL2 shRNA expression). Cells with low expression levels of the different constructs were used in all the studies.

**Cell line transfections.** For live imaging and immunocytochemistry experiments, Cos7 cells were transfected using SuperFect (Qiagen, 301305) according to man-ufacturer's instructions. Briefly, DNA (0.5 µg/per construct for overnight expres-sion) was mixed with 7.5 µL of SuperFect in 75 µL DMEM and incubated for 10 min, combined with 400 µL DMEM/FCS plating medium and added to cells overnight. Next day transfection medium was replaced with fresh medium at least 2 h before imaging or fixation.

For pull down experiments HEK cells were transfected using PEI (Polyethylenimine HCl MAX Linear MW 40000 (PolySciences, 24765–2)). Briefly, 30 µL PEI (1 µg/µL) was mixed with 10 µg DNA in 1000 µL Ham's F10 and incubated for 30 min. Transfection medium was added to cells in 10 cm plate overnight. Next day transfection medium was replaced with fresh medium and cells were collected after total of 48 h post transfection.

**Immunofluorescence fixation and staining.** For morphology quantifications, membrane recruitment assay and protein colocalizations, neurons, Cos7 cells and HeLa cells were fixed in 4% paraformaldehyde with 0.2% Triton X-100 in PHEM buffer for 15 min[16]. For colocalization of RFP-APC2 clusters with tubulin bundles, neurons were first extracted for 10 min with 0.04% Triton X-100 in PHEM buffer supplemented with 10 µM taxol (Sigma-Aldrich, T7402) and then fixed for 15 min in 4% paraformaldehyde with 0.2% glutaraldehyde in PHEM buffer. Colocaliza-tions with EB3 were done in cells fixed in 100% Methanol at −20 °C for 10 min. Human iPSC derived neurons were fixed in 100% Methanol at −20 °C for 2 min, followed by 4% paraformaldehyde with 4% sucrose. Following fixation, cells were washed three times 5 min in PBS. Cells fixed with glutaraldehyde were then quenched with NaBH$_4$ 10 mg/mL in PBS twice for 10 min and washed three times in PBS again. Cells were then blocked in 10 mg/mL Albumin, 10% normal goat serum in PBS for 1 h and incubated with primary antibody at 4 °C overnight, washed three times in PBS and incubated with secondary antibody at 37 °C for 1 h. Coverslips were washed again three times and mounted with Mowiol. SiR-Actin was used post fixation alongside secondary antibodies (0.3 µM per coverslip).

**Laser Scanning Confocal and STED microscopy.** Confocal and gated STED imaging was performed with Leica TCS SP8 STED 3× microscope using HC PL APO ×100/1.4 oil STED WHITE objective. All tubulin, SiR-Actin, and EB3 were imaged with STED microscopy. During imaging, secondary probes Alexa 488, Alexa 647 or SiR-Actin were excited with the 488 or 633 nm wavelength of pulsed white laser (80 MHz) and depleted with CW 592 nm or 775 nm STED lasers respectively. Images were acquired in 2D STED mode. Depletion laser power was equal to 70% of maximum power and we used an internal Leica GaAsP HyD hybrid detector with a time gate of $1 \le t_g \le 8$ ns. Confocal imaging was used on all RFP or GFP fused expressing constructs and was performed on the same setup using white laser and standard excitation and emission settings from LAS X controlling software library.

**Live-cell imaging**. Time-lapse live-cell imaging was done using spinning-disk microscopy. An inverted research microscope Nikon Eclipse Ti-E (Nikon) was equipped with the perfect focus system (PFS) (Nikon), Plan Apo VC ×100 NA 1.40 oil objective (Nikon), CSU-X1-A1 Spinning Disc (Yokogawa), and Photometrics Evolve 512 EMCCD camera (Roper Scientific), controlled by MetaMorph 7.7 software (Molecular Devices). Images were projected onto the camera chip with intermediate lens 2.0 × (Edmund Optics) at a magnification of 0.067 µm/pixel. Coverslips were mounted into Ludin-style perfusion chambers and maintained at 37 °C and 5% $CO_2$ using stage top incubator INUBG2E-ZILCS (Tokai Hit) for the duration of imaging. The microscope was equipped with a custom-ordered illuminator (Nikon, MEY10021) modified by Roper Scientific France/PICT-IBiSA, Institut Curie. A 491-nm (100 mW) Calypso (Cobolt) and a 561-nm (100 mW) Jive (Cobolt) lasers were used for excitation. The spinning disk was equipped with a 405-491-561 triple-band mirror and GFP, mCherry, and GFP/mCherry emission filters (Chroma). For simultaneous imaging of green and red fluorescence we used an ET-mCherry/GFP filter set (59022, Chroma) together with the DualView (DV2, Roper) equipped with the dichroic filter 565dcxr (Chroma) and HQ530/30 m emission filter (Chroma).

Data were analyzed in ImageJ Fiji. Maximum projections were generated for display purposes. Data were analyzed from kymographs generated using KymoResliceWide plug-in and manual counting. Color-coded maximum projections for anterograde and retrograde + TIP comets were generated using Correlescence v.0.0.4 plug-in. Color-coded kymographs for anterograde and retrograde + TIP comets were generated using for Split-antero-retro macro. Correlescence v.0.0.4 plug-in, KymoReslice Wide and Split-antero-retro macro were generously provided by Dr. E. Katrukha (Utrecht University) and are available on GitHub.

**Analysis of neuronal morphology**. ImageJ Fiji was used to create a mask of concentric circles every 15 µm starting at the center of cell body for Sholl analysis. Every crossing of MARCKS-BFP expressing neuron was manually counted at each consecutive circle. Dendrite thickness was measured in ImageJ at the base of each primary dendrite within 5 µm of the cell body. Length of spines was also measured in ImageJ and classified into three categories—mushroom, filopodia and stubby based on shape (see Fig. 1 legend for more details).

**qPCR**. Primary hippocampal neurons (200,000 per condition) were electroporated with either puro-control or puro-APC2 shRNA using NEPA GENE according to the manufacturer's instructions in Opti-MEM (Gibco) and plated in plating medium in 12 well plates coated with poly-L-lysine (37.5 µg/mL) and laminin (1.25 µg/mL). Cells were maintained at 37 °C in 5% CO2 before collection. After 1 DIV, neurons were treated with 0.5 µg/mL puromycin for 48 h. In parallel, puromycin was added to neurons that underwent a mock transfection with no DNA to check for selectivity of the puro cassette. After a total of 3 DIV neurons were washed once in pre-warmed PBS and scraped in Trizol (Thermo Fisher, 15596026). Treatment of mock transfected neurons with puromycin resulted in 100% cell death as expected, while experimental conditions contained at least 50% healthy neurons post puromycin treatment. Collected samples underwent RNA isolation using RNeasy mini kit (Qiagen, 74104) according to manufacturer's instructions. cDNA was synthesized from the RNA samples using Superscript IV First-Strand Synthesis System (Thermo Fisher, 18091050) according to manufacturer's instructions. The qPCR reaction was performed using the LightCycler (Roche) and the Fast Start DNA Master PLUS SYBR green I kit (Roche). Primers used for APC2 were (5′-CTTGAAGGCGGAGAACACTC) and (3′-GATACCAGCACTCTCGC CTC). Primers used for GAPDH were (5′-TGCCCCCATGTTTGTGATG) and (3′-TGTGGTCATGAGCCCTTCC). Gene expression was calculated as normalized ratio and normalized to reference gene GAPDH using the ΔΔCt method.

**Polarity index**. A 10 µm long region of max. two dendrites per neuron was selected and measured for raw intensity of fluorescence signal. A 10 µm long region of the axon was selected past the TRIM46 immunostained region and measured for raw intensity of fluorescence signal. Background values were subtracted and values were corrected for α-tubulin. Polarity index (PI) was calculated using the following formula: $PI = (Id - Ia)/(Id + Ia)$, where Id is the mean dendrite intensity and Ia is the mean axonal intensity. Non-polarized proteins present a $PI = 0$ ($Id = Ia$), whereas $PI > 0$ or $PI \leq 0$ indicates polarization toward dendrites or axons, respectively. All measurements were done in ImageJ.

**Drug treatments**. Neurons were treated with either 10 µM Nocodazole (Sigma-Aldrich, M1404) or equal volume of DMSO for 90 min during imaging. Cos7 cells were treated with either 1 µM Rapalog (Takara, 635057) or equal volume of 100% ethanol for 40 min during live-imaging and for 1 h for fixed cell fluorescence studies.

**FRAP experiments**. For quantitative fluorescence recovery after photobleaching (FRAP) experiments, neurons were imaged on the spinning disk microscope described above. FRAP experiments were performed using the ILas2 system (Roper Scientific). Regions of primary dendrite were photobleached with high laser power (10 times at 20% laser power with a Vortran Stradus 405 nm (100 mW) laser) and imaged every 30 s for fluorescence recovery for a period of 10 min. For FRAP

analysis, the mean intensity of the bleached area was corrected for background values, as well as the bleaching that occurred during image acquisition. Data were normalized to control fluorescence averaged over 1–3 initial frames before bleaching and stated as 100% intensity. Average curves were obtained and represented.

Time-lapse of nocodazole treated RFP-APC2 expressing neurons were analyzed in the same manner: background values were subtracted and data were corrected for bleaching.

**Photoactivation experiments**. Photoactivation of PA-GFP-α-Tubulin was performed on the FRAP setup and achieved by scanning the desired region (5–10 µm long section of proximal dendrite) three times at 12% laser power with a Vortran Stradus 405 nm (100 mW) laser. Photoactivated region was imaged every 30 min for 6 h. This time-lapse was determined optimal by testing imaging frequency from every 1 min to 30 min for 1 h up to 18 h. ROI were considered to translocate when whole region moved unanimously towards the cell body throughout the duration of imaging. ROI was considered to undergo microtubule bundle sliding when bundles were observed sliding out of the ROI towards and away from cell body, while the middle of the ROI primarily remained stationary. Changes in distance were measured by tracing borders of the ROI at time 0, 3 h, and 6 h.

**Photo-ablation experiments**. A Teem Photonics 355 nm Q-switched pulsed laser was used for photo-ablation on the described spinning disk microscope using S Fluor ×100, 0.5–1.3 NA oil-immersion objective (Nikon)[10]. For microtubule orientation experiments, microtubules were photo-ablated in proximal dendrites and imaged every 0.5 s for 3 min. For branch point guidance experiments, microtubules were photo-ablated 10 µm prior to 1st branch point and 10 µm post at both branches. Cells displayed no significant distress, such as blebbing or significant changes in morphology for the duration of imaging. Same settings were used for microtubule laser photo-ablation in Cos7 cells.

**Motor-PAINT sample preparation**. Hippocampal neurons were transfected as described above with TagRFP-α-tubulin and either control or APC2 shRNA on DIV11 and prepared for imaging on DIV15. To prepare cellular microtubule cytoskeletons for the kinesin motility assays, the cytoplasm of hippocampal neurons was extracted for 1 min with extraction buffer (1 M sucrose, 0.15% Triton X-100 in PEM80 buffer) at 37 °C. Subsequently, an equal amount of fixation buffer (2% PFA in PEM80 at 37 °C) was added and the solution was gently mixed by pipetting for 1 min. The extraction and fixation buffer were then replaced by washing solution (100 nM Paclitaxel in PEM80 buffer at 37 °C) for 1 min. Samples were washed three times, 1 min each, and placed into imaging buffer (1.7% glucose, 185 µg/ml glucose oxidase, 40 µg/ml catalase, 5 mM ATP, 1 mM DTT, 100 mM Paclitaxel in PEM80 buffer at 37 °C)[41].

**Motor-PAINT imaging and analysis**. Fixed samples were imaged on a Nikon Ti-E microscope equipped with a 100 × Apo TIRF oil immersion objective (NA. 1.49) and Perfect Focus System 3[41]. TagRFP-α-tubulin expressing cells were selected for imaging and 1 µl of 30 nM DmKHC(1–421)-GFP-His was added above the location of acquisition. Recombinant kinesin was supplemented during imaging to increase the number of localizations of motile kinesins when diffusion and photobleaching impeded imaging. Excitation was achieved via a custom illumination pathway starting with a Lighthub-6 (Omicron) containing a 638 nm laser (BrixX 500 mW multimode, Omicron), a 488 nm laser (Luxx 200 mW, Omicron) and using an optical configuration that allowed tuning the angle of incidence. Total internal reflection was used for illumination. Microscope was equipped with a quad-band polychroic mirror (ZT405/488/561/640rpc, Chroma), a quad-band emission filter (ZET405/488/561/640 m, Chroma), an additional single-band emission filter (ET525/50 m for GFP emission, Chroma), and detected using a sCMOS camera (Hamamatsu Flash 4.0v2). For analysis, 10,000–20,000 frames were acquired at 10 Hz using stream acquisition. Microtubule orientations along 10 µm of proximal dendrite were analyzed by separating localizations corresponding to minus and plus-end-out microtubules according to method previously described[41].

**Colocalization quantifications**. Line scans were performed in ImageJ and graphs were prepared in Excel. Colocalization coefficients were obtained by Coloc 2 plug-in in ImageJ from background subtracted ROIs. For RFP-APC2 colocalization with acetylated and tyrosinated tubulin, thresholded Manders' coefficient was calculated from one 5 × 5 µm ROI per neuronal soma from pre-extracted and PFA fixed neurons. For C-terminus colocalization with microtubules in Cos7 cells, Pearson's coefficient was calculated from max three 10 × 10 µm areas per cell. For colocalization of DYNLL2-GFP and APC2 truncations in Cos7 cells Manders' coefficient M2 was measured of red channel over the green from PFA fixed cells. Colocalization of DYNLL2-GFP and RFP-APC2 was measured from live cells. Max two 10 × 10 µm ROIs were measured per cell. For HeLa colocalizations Pearson's correlation was used with manual thresholding on max two 10 × 10 µm areas per cell. For LC8 recruitment to RFP-APC2 in neurons Pearson's correlation was used with manual thresholding on 15 µm long areas of max two dendrites per PFA fixed neuron.

**Inducible membrane recruitment assay**. Cos7 cells were co-transfected with FKBP-eGFP-CAAX and FRB-RFP-APC2-C or FRB-RFP-APC2-C1 or FRB-RFP-APC2-C2 and imaged after 24 h of expression. Coverslips were mounted into Ludin chambers and imaged for 3 min with frames taken every 0.5 s prior to treatment. Then either control (100% ethanol) or rapalog was added and cells were imaged for 40 min with frames taken every 30 s. At the end of the 40 min treatment another 3 min movie was made to be able to compare before and after effects of the rapalog driven recruitment of APC2 C-terminus truncations to the membrane. For quantification, ROI was traced in ImageJ and fluorescence intensity of FKBP-eGFP-CAAX was determined after 40 min treatment. Percentage of fluorescence arising from bundles localized to FKBP-eGFP-CAAX was measured by subtracting mean fluorescence from non-bundled region of FKBP-eGFP-CAAX from total mean fluorescence intensity of the cell.

**Biotin-streptavidin pull-down assay**. HEK293T cells were transfected with either a BioGFP-fusion construct (pull down bait) together with a plasmid encoding the biotin ligase BirA or RFP-fusion constructs (pull down prey). After 48 h cells were lysed in lysis buffer (50 mM Tris-HCl pH 7.5, 100 mM NaCl, 0.5% Triton X-100, 5 mM $MgCl_2$ and protease inhibitors (Roche)). Cell lysates were centrifuged at 16,363 x g for 15 min and the supernatants were mixed according to conditions 1:1 BioGFP-fusion lysate to RFP-fusion lysate. Dynabeads M-280 streptavidin (Invitrogen) were pre-treated with blocking buffer for 1 h at RT (20 mM Tris-HCl pH 7.5, 150 mM KCl, 0.2 μg/μL chicken egg albumin) and then incubated with the lysate mixtures at 4 °C for 90 min. Beads were separated using a magnet (Invitrogen) and washed 5 times in wash buffer (20 mM TrisHCl pH 7.5, 150 mM KCl, 0.1% Triton X-100). Input samples were incubated on ice for the duration of procedure. For protein elution, input samples and the beads were boiled in SDS sample buffer for 8 min, separated by magnet, and supernatants were run on a 6–16% gradient SDS-PAGE gels in a Mini-PROTEAN cassette (Bio-Rad). Western blotting was performed on nitrocellulose membrane using semi-dry transfer (Bio-Rad). Blots were blocked with 3% Bovine Serum Albumin in 0.05% Tween-20 in TBS for 1 h at RT and incubated with primary antibodies at 4 °C overnight. Blots were washed with 0.05% Tween 20 in TBS three times for 10 min at RT and incubated with either anti-rabbit or anti-mouse IRdye antibodies of 1 h at RT. Blots were washed again three times at RT and imaged using Odyssey Infrared Imaging system (LI-COR Biosciences).

**Human iPSC-derived cortical neuron cultures**. Commercial human induced Pluripotent Stem Cell (hiPSC)-derived Neural Stem Cells (NSCs) from a female control donor (Axol Bioscience, ax0016) were plated in Neural Plating-XF Medium (Axol Bioscience, ax0033) on ReadySet + SureBond-coated (Axol Bioscience, ax0041) glass coverslips at a density of 50,000–200,000 cells/well of 24-well plates. The next day, medium was fully replaced with Neural Expansion-XF Medium (Axol Bioscience, ax0030). After 24 h, neuronal differentiation was induced by switching to Neural Differentiation-XF Medium (Axol Bioscience, ax0034), which was refreshed twice a week by replacing half of the medium. When NSCs were differentiated after 3–6 days, medium was fully replaced by Neural Maintenance-XF Medium (Axol Bioscience, ax0032) supplemented with 2 μM Ara-C (Sigma, C1786). Medium of differentiated NSCs was changed twice a week by replacing half of the Neural Maintenance-XF Medium.

**Tubulin level ratios**. Whole Cos7 cells expressing either RFP, RFP-APC2 or RFP-APC2-C were measured for raw fluorescence intensity of acetylated tubulin and ratioed against α-tubulin immunofluorescence levels. All samples and repeated experiments were stained using same antibody concentrations.

**Data analysis and quantification**. Data were collected and analyzed from 2 to 4 independent experiments in Excel. No specific strategy for randomization and/or stratification was employed. The studies were blind in data analysis. No samples were excluded from analysis. Figures were generated in Illustrator CC.

**Statistical analysis**. All statistical details of experiments, including the definitions and exact values of n, and statistical tests performed, can be found in figures and figure legends. Data processing and statistical analysis were done in Excel and SPSS. Significance was defined as: "n.s." not significant or $p > 0.05$, "∗" for $p \leq 0.05$. All data were checked for normality by Shapiro–Wilk test and homogeneity of variances by Levene's test prior to choosing appropriate comparison test. Statistical tests include: One sample t-test, Mann–Whitney U test, Wilcoxon signed-rank test, Independent samples t-test, One-Way Anova with Tukey post-hoc, Two-Way Anova with Tukey post-hoc, Kruskal–Wallis Test with Dunn's post-hoc. Tests were two-tailed.

**Data availability**. The data that support the findings of this study are available from the corresponding author upon request.

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

## Acknowledgements

We thank Dr. Jeroen Pasterkamp and Marina de Wit for their assistance with qPCR experiments and providing access to LightCycler. We thank Dr. Eugene Katrukha for providing access to Correlescence v.0.0.4 plug-in and Split-antero-retro macro and Cao Yujie for advice with cloning. We are grateful to Dr. Raymond Poot for pSuper.puro EV and Dr. Eva Wenzel for YFP-hAPC2. This work was supported by the Netherlands Organization for Scientific Research (NWO-ALW-VICI, CCH) and the European Research Council (ERC) (ERC-consolidator, CCH).

## Author contributions

O.I.K. conceptualized the project, performed most of the experiments, analyzed data and wrote the manuscript. P.S. generated several constructs and conceptualized the project. D.W. assisted with qPCR and pull-down experiments. R.P.T. performed motor-PAINT experiments and analyzed related data. F.W.L. and S.P. cultured and differentiated the neuronal iPS cells. L.C.K. provided project feedback. C.C.H. supervised the study and wrote the manuscript.

## Additional information

**Competing interests:** The authors declare no competing interests.

