## [Peer Review File · Nature Communications]

Reviewers' comments:

Reviewer #2 (Remarks to the Author):

The manuscript by Kahn et al. examines the role of APC2 in dendrite development of hippocampal neurons and identifies a potential mechanism by which APC2 could be regulating cytoskeletal organization in neurons. While the role of APC2 in dendrites of mammalian neurons (lot of previous work done in Drosophila neurons) in itself would be of interest to the cytoskeleton field, some of the experiments are difficult to interpret, making it hard for the reader to place the effects of APC2 in a physiological context. I have several specific comments/concerns about the manuscript in its current form, outlined below.

1. The authors report that full length RFP-APC2 localizes exclusively to dendrites (Figure 2). Previous work from the Noda lab has shown that APC2 localizes to growth cones and axonal shafts. Is anything known about the localization of APC2 during the course of development? Can the authors examine the localization of APC2 in developing neurons? Does APC2 localization change from axons to dendrites during the course of development? Or could it be an issue of very low expression levels that the authors do not see any APC2 in axons? If expression levels are an issue, can the authors stain for endogenous APC2 in both young, developing neurons (<7 DIV) and fully differentiated neurons.

2. In Figure 2D, the authors show images of APC2 puncta with acetylated vs tyrosinated tubulin and report that the majority of puncta co-localize with tyrosinated tubulin. I am not sure one can definitively say that a protein binds to acetylated or tyrosinated tubulin using regular confocal microscopy in dendrites. Also, these images are not very clear and if anything, in the merged images, there seem to be more co-localization of the APC2 clusters with acetylated tubulin.

3a. All the data on the effect of APC2 on the dynamics of microtubules (MTs) is very confusing at the moment (Figure 3 and Supplementary figure 2). The authors report differential effects of APC2 on plus-end out microtubules vs minus-end out microtubules. While in the text, the authors state the results their data demonstrate, it is hard to understand the exact effects of APC2 given the sometimes contradictory nature of these results: for example, both depletion of APC2 and over expression of APC2 decreases the microtubule assembly track of retrograde comets (Supplementary Figure 2B, D). While the data could be statistically significant, what does this mean biologically? It is unclear to me how APC2 could be specifically distinguishing the two sets of microtubules and further have differential effects on them.

3b. I am also curious about the reported velocities of the comets. 0.6-0.8 $\mu\text{m/s}$ seems quite high. Is there any reason why the authors see velocities largely different from other published papers (including their own Yau et al., 2016) that have reported 0.1-0.2 $\mu\text{m/s}$ (see Stepanova et al 2003, Sweet et al., 2011)

4. The authors use PA-GFP tubulin to examine overall translocation of the microtubule cytoskeleton and demonstrate that loss of APC2 causes increased translocation of MTs. How does this finding tie in with the effect of APC2 on the dynamics of minus-end out MTs?

Without any explanation for what the authors think this result means, it is hard to interpret this result in the context of the mixed MT orientation in dendrites. Further, what would the result for this assay be when done in the presence of over-expressed APC2.

5. While the authors convincingly demonstrate that the C-terminus of APC2 is the MT binding site, full length APC2 is found only in punctae or clusters along microtubules. Given this, it is hard to interpret the results of some of the truncated constructs.

(i) For example, in Figure 6A, the authors measure the time it takes for MTs to polymerize when bundled by the C-terminus of APC2 vs TRIM46 vs PRC1. The data demonstrate that APC2 C-terminus slows down polymerization, implying it slows down MT dynamics. This data is in stark contrast to the findings in neurons where loss of APC2 causes decreased MT dynamics.

(ii) The C-terminus of APC2 also has an SxLP motif which binds to MT tips but the full length APC2 does not seem to bind to MT tips. Does endogenous APC2 exist in two forms – one binding to the MT tips and one binding to MT lattice? And can it switch from one form to another? If so, what would the switch be? Can the authors do single molecule motility assays with purified APC2 to nail down the specific effects of APC2 on MT dynamics?

6. The authors propose that APC2 clustering is driven by LC8 binding (Figure 7). The data in Figure 7E indicates that the C-terminus of APC2 recruits DYNLL2/LC8 to MTs. Is this true for full length APC2 as well? While the data in Figure 7A indicates that DYNLL2 co-localizes with APC2, what is the localization of DYNLL2 in neurons without any over-expression of APC2. Would it be cytoplasmic or punctae? The authors also go on to identify the specific region in the C-terminus that is required for the binding of LC8. The authors then express the mutated APC2 in neurons and conclude that LC8 binding is required for the proper localization of APC2. However, previous reports have shown that mutation in the C-terminus of APC2 acts as a loss-of-function. Hence, the mislocalization in this case could be due to this. Can the authors knock down endogenous LC8 and then observe localization of APC2? This would eliminate the use of mutant constructs and will also strengthen their claim about LC8 binding determining the localization of APC2 in neurons.

Reviewer #3 (Remarks to the Author):

The study titled "APC controls dendrite development by promoting microtubule dynamics" is generally appropriate for nature communications and should be acceptable with significant revisions. The authors show a novel phenotype for APC2 by illustrating that APC2 depletion causes a significant reduction in the number of dendritic spines in cultured rat hippocampal neurons and fewer retrograde microtubule comets. The authors then find that expression of an RFP-tagged APC2 localizes to microtubules in dendrites, which depend on elements in the C-terminal portion of APC, as loss of the C-terminus mobilizes APC2 puncta and reduces colocalization with microtubules. More careful investigation of the C-terminus reveals varying requirements of certain C-terminal elements, with the distal portion of APC2 binding microtubule +ends, while an expanded S-domain is responsible for binding microtubule bundles. Furthermore, this patterning of APC2 along microtubules is dependent on a domain

in APC2 that also interacts with the dynein complex member LC8. However, the authors make some claims that are not sufficiently supported and rely on reagents that are not entirely validated, and therefore several major issues remain to be addressed before a recommendation for publication can be made.

Major issues:

1. The sole use of RFP-APC2 overexpression constructs to make conclusions about APC2 localization limits the significance of these findings. Given that the localization and behavior of overexpressed and tagged constructs in relation to endogenous protein are not always reliable, RFP-APC2 should be validated by showing that expression of RFP-APC is capable of rescuing APC2 knockdown in terms of dendrite morphology. Additionally, localization of RFP-APC2 should be validated by comparing it to the localization of endogenous APC2.

2. The title of this study is "APC2 controls dendrite development by promoting microtubule dynamics". However, the direct connection between dendrite morphology and the observed changes in microtubule dynamics was not clearly established. To support this claim, the authors should confirm that the APC2 mutants, which alter the localization and interaction with microtubules, have some effect on dendrite morphology. By determining if a particular mutant is capable of rescuing the APC2 knockdown phenotype with regards to dendrite morphology, the authors could make the argument that association between APC2 and microtubules is necessary for proper dendrite development.

Specific/ minor points and experimental controls:

Figure 1

- The authors only quantify dendritic spines that possess both Homer and Bassoon staining and note a significant reduction in dendritic spine quantity with APC2 knockdown. This allows for the possibility that APC2 knockdown may be causing reduced levels of Homer and Bassoon resulting in number of dendritic spines being inappropriately counted. A control for Homer and Bassoon levels should be done for APC2 shRNA vs control shRNA.

Figure 2

- Since APC2-RFP is being overexpressed, it is important show quantifications of any effects in dendrite or dendritic spine morphology with this background.

- In Fig2D, the authors claim that the majority of RFP-APC2 clusters colocalize with tyrosinated microtubule bundles. This should be demonstrated with quantification especially since determination based on the provided images alone is not clear.

- A control confirming that nocodazole and SpvB treatment is sufficiently disrupting the cytoskeleton should be included.

Figure 3 and Sup Figure 2

- In Fig3 G,H, quantifications for retrograde comets are given as percents of total comets. This is not ideal since a difference in percent retrograde comets could be due to changes in the actual number of retrograde comets or due to changes in anterograde comets. Presenting the data as it is in Fig.2E would be more helpful.

- Comparing MT assembly velocity in samples with addition of control shRNA in FigS2B with

addition of just RFP in FigS2D reveals a substantial difference in two samples that should be similar. Given this discrepancy, showing MT assembly velocity in a sample where neither control shRNA or RFP is added would be helpful.

Figure 4

- It is unclear in Fig4B what the quantification from Fig4C represent. Also, explicit indication should be given for what the dotted lines and arrows represent in B.

Figure 5 and Sup Figure 3

- The legend on the graph in Fig6B (Area of cell %) is unclear given the description in the figure legend (Representative image of area of microtubules...)
- RFP-APC and RFP-APC2-C2 should be quantified with the others in Fig2B.
- As mentioned by the authors, level of RFP-APC2 expression can have profound effects on APC2-RFP localization. A control that confirms similar levels of expression for all mutants shown in Fig5 should be included.
- Colocalization quantifications between RFP-APC2 fragments and SiR-Actin should be included.

Figure 6

- The authors state "These results suggest that RFP-APC2-C may indeed prefer freshly assembled dynamic microtubules and stabilize them upon binding, which in turn increases their acetylation level with time". To conclude this, an experiment looking at the relative amounts of acetylated tubulin in samples with overexpressed RFP-APC-C and RFP-APC vs control needs to be done.
- It is unclear whether the kymographs in Fig6E represent pre or post rapalog treatment.
- A key point argued in Fig6 is that the RFP-APC-C2 fragment is capable of catching growing microtubule plus ends and stabilizing them when tethered to the cell membrane. This point would be better supported if a clear way of quantifying the images in Fig6G in terms of immobilization of MTs was included. These quantifications should include comparisons of the ability of APC-C, APC-C1 and APC-C2 to immobilize MTs.

Figure 7

- The authors state "RFP-APC2 clusters in neurons significantly colocalize with GFP-DYNLL2 and to a lesser degree with GFP-DYNLL1", but no quantifications are shown.
- Fig7G should include quantifications of colocalizations between RFP-APC2 and GFP-DYNLL2.
- To establish that DYNLL2 is responsible for proper localization of RFP-APC2, DYNLL2

should be depleted and localization of RFP-APC2 should be assessed.

Reviewer #4 (Remarks to the Author):

This is an exciting manuscript, executed beautifully, and presented magnificently. The question addressed here is important to neuroscientists interested in how neurons regulate the various details of their microtubule arrays. The question might be a tad narrow for a general audience, but the work is so masterfully done that I believe cell biologists will find insight relevant to many cell types in this work. The issue is how the minus-end-out microtubules in the vertebrate dendrite have an overall different level of stability than the plus-end-out microtubules, with the conclusion being that APC2 clusters keep the plus ends of the minus-end-out-microtubules more dynamic. There's a great deal of data in the paper, certainly no need for more, with a number of different molecular factors and microtubule behaviors studied.

We thank the three reviewers for their positive assessment and the constructive critiques. In order to address all the comments and include reviewers' suggestions we have made numerous changes in the figures and text of the manuscript. Find below our detailed response to all of the concerns made by the reviewers and the changes made to the original version of the manuscript.

***** Reviewer #2 *****

The manuscript by Kahn et al. examines the role of APC2 in dendrite development of hippocampal neurons and identifies a potential mechanism by which APC2 could be regulating cytoskeletal organization in neurons. While the role of APC2 in dendrites of mammalian neurons (lot of previous work done in *Drosophila* neurons) in itself would be of interest to the cytoskeleton field, some of the experiments are difficult to interpret, making it hard for the reader to place the effects of APC2 in a physiological context. I have several specific comments/concerns about the manuscript in its current form, outlined below.

Thank you for finding our work of interest to the cytoskeleton field. We appreciate your insightful comments and suggestions. We have added several experiments to our manuscript that address the comments below, improved our data representation by adding schematics and two new movies, and amended the text of the manuscript to clarify our data explanations and conclusions.

1. The authors report that full length RFP-APC2 localizes exclusively to dendrites (Figure 2). Previous work from the Noda lab has shown that APC2 localizes to growth cones and axonal shafts. Is anything known about the localization of APC2 during the course of development?

*Indeed, the Noda lab has shown that APC2 is preferentially expressed in the nervous system from early developmental stages through adulthood and is distributed along microtubules and actin in growth cones, as well as axon shafts of chick retinal axons and cerebellar granule cells, all early in development. The Rolls lab has shown that APC2 in vivo in *Drosophila* central neurons and dendritic arborization neurons is localized to cell bodies, dendrites, and some proximal axons, but not distal axons. In *Drosophila* neurons APC2 is one of the most polarized markers that distinguishes *Drosophila* dendrites from axons. Within dendrites it is very specifically localized to dendrite branch points where APC2 plays a role in controlling microtubule polarity. Here, we investigated function of APC2 in fully differentiated hippocampal rat neurons and used human construct for localization studies. We have added text to the first paragraph of Results section "APC2 clusters localize to microtubule bundles in dendrites" of the manuscript to address this point.*

Can the authors examine the localization of APC2 in developing neurons? Does APC2 localization change from axons to dendrites during the course of development?

Thank you for this suggestion. We have now examined localization of RFP-APC2 in hippocampal neurons at DIV 3 (young), DIV 7 (developing) and DIV 15 (fully differentiated) and quantified the distribution by measuring the polarity index of the protein. We found that RFP-APC2 localizes throughout the neuron with some enrichment in the longest neurite at DIV 3 prior to axon initial segment development. RFP-APC2 relocalizes to dendrites after the axon initial segment development; both at DIV 7 and at DIV 15. This data is now in Figure 2b and the first paragraph of Results section "APC2 clusters localize to microtubule bundles in dendrites".

Or could it be an issue of very low expression levels that the authors do not see any APC2 in axons? If expression levels are an issue, can the authors stain for endogenous APC2 in both young, developing neurons (<7 DIV) and fully differentiated neurons.

We did not find expression levels to be an issue in localizing RFP-APC2. Our data in Figure 2b confirms that RFP-APC2 localizes to axons of rat hippocampal neurons early in development and is restricted to dendrites upon full differentiation, in line with published work from Noda and Rolls labs. We have not stained for endogenous APC2 in the original manuscript, because the commercial antibodies we have acquired, including ones from published works, do not successfully recognize rat APC2 (via immunocytochemistry and Western Blots: the antibodies were unable to recognize correct sized band and/or depletion of APC2). Since all available antibodies are raised against the human peptides, we concluded that the antibodies must recognize human APC2 and not rat. In order to address this comment of the reviewer, we have now confirmed this by successfully staining our human RFP-APC2 expressed in rat neurons in Supplementary Figure 1b. To confirm that our expressed APC2 localizes similarly to endogenous, we used 2 of the commercial antibodies (one against the C-terminus and another against the middle of APC2) to stain human iPSC-derived neurons and confirmed that endogenous APC2 localizes in clusters in MAP2 positive processes (Supplementary Figure 1b).

2. In Figure 2D, the authors show images of APC2 puncta with acetylated vs tyrosinated tubulin and report that the majority of puncta co-localize with tyrosinated tubulin. I am not sure one can definitively say that a protein binds to acetylated or tyrosinated tubulin using regular confocal microscopy in dendrites. Also, these images are not very clear and if anything, in the merged images, there seem to be more co-localization of the APC2 clusters with acetylated tubulin.

We agree with the reviewer, that we cannot definitively say that APC2 binds to either acetylated or tyrosinated tubulin using regular confocal microscopy in dendrites. We have repeated this experiment and moved our quantifications to the cell body of the neurons, where microtubules are more spaced out. We quantified colocalization of RFP-APC2 with bundles of microtubules that were stained for tyrosinated or acetylated tubulin in neurons that were extracted prior to fixation. We found no significant difference in colocalization Manders' coefficient values (0.42 ± 0.03 for acetylated tubulin and 0.39 ± 0.03 for tyrosinated tubulin, $p=0.484$, Independent samples T-Test). We have added this data to Figure legend 2e and adjusted our conclusions in the manuscript relating to this figure panel (now Figure 2e) in the second paragraph of Results section "APC2 clusters localize to microtubule bundles in dendrites" and in the Discussion.

3a. All the data on the effect of APC2 on the dynamics of microtubules (MTs) is very confusing at the moment (Figure 3 and Supplementary figure 2). The authors report differential effects of APC2 on plus-end out microtubules vs minus-end out microtubules. While in the text, the authors state the results their data demonstrate, it is hard to understand the exact effects of APC2 given the sometimes contradictory nature of these results: for example, both depletion of APC2 and over expression of APC2 decreases the microtubule assembly track of retrograde comets (Supplementary Figure 2B, D). While the data could be statistically significant, what does this mean biologically? It is unclear to me how APC2 could be specifically distinguishing the two sets of microtubules and further have differential effects on them.

Thank you for pointing out that the way the data was presented and explained is confusing. We have reassessed biological relevance of APC2 overexpression effect on microtubule dynamics and agree with the reviewer that the effect on microtubules in this overexpression case is difficult to interpret and lacks biological relevance. APC2 overexpression may have aberrant effects unrelated to its normal function. Therefore, we removed the overexpression experiment panels from Supplementary Figure 2 and made appropriate adjustments in the text. We have retained APC2 expressing experiments in a depleted endogenous protein background (rescue context) and experiments where expression is used solely for localization. We have also substantially improved our explanation of the results on microtubule dynamics data in the Results section “APC2 depletion alters minus-end-out microtubule dynamics in dendrites” and added clarifications on how we propose APC2 functions into our discussion.

3b. I am also curious about the reported velocities of the comets. 0.6-0.8 $\mu\text{m/s}$ seems quite high. Is there any reason why the authors see velocities largely different from other published papers (including their own Yau et al., 2016) that have reported 0.1-0.2 $\mu\text{m/s}$ (see Stepanova et al 2003, Sweet et al., 2011)

We are grateful to the reviewer for catching this discrepancy. In our original calculation the frame rate was not adjusted appropriately to the 2 frames per second that the movies were taken at. Once we fixed the frame rate, the calculations readjusted to the reported rates of 0.1-0.2 $\mu\text{m/sec}$. The graph is now fixed in new Supplementary Figure 2c.

4. The authors use PA-GFP tubulin to examine overall translocation of the microtubule cytoskeleton and demonstrate that loss of APC2 causes increased translocation of MTs. How does this finding tie in with the effect of APC2 on the dynamics of minus-end out MTs? Without any explanation for what the authors think this result means, it is hard to interpret this result in the context of the mixed MT orientation in dendrites. Further, what would the result for this assay be when done in the presence of over-expressed APC2.

Thank you for catching lack of an explanation in the text. We have now added our conclusions based on the PA-GFP tubulin experiment data into the manuscript in the first paragraph of Results section “APC2 effect on global organization of microtubule polarity in dendrites”. We have also adjusted the figure to add clarity and added a movie to show whole microtubule cytoskeleton translocation related to Figure 4c (Movie 1). We have also done the suggested experiment using over-expressed APC2, however ran into issues with significant increase in RFP-APC2 expression in the neurons over the 6 hours of imaging. We concluded that obtained data was of no biological relevance and decided on not including it in the manuscript.

5. While the authors convincingly demonstrate that the C-terminus of APC2 is the MT binding site, full length APC2 is found only in punctae or clusters along microtubules. Given this, it is hard to interpret the results of some of the truncated constructs.

(i) For example, in Figure 6A, the authors measure the time it takes for MTs to polymerize when bundled by the C-terminus of APC2 vs TRIM46 vs PRC1. The data demonstrate that APC2 C-terminus slows down polymerization, implying it slows down MT dynamics. This data is in stark contrast to the findings in neurons where loss of APC2 causes decreased MT dynamics.

These two experiments address different microtubule populations in different conditions and cell types. Figure 6a addresses effect of RFP-APC2-C, which is an actively bundling microtubule binding truncation construct, on forced repolymerization of the microtubule bundle induced by photoablation. The experiment is meant to examine the difference of function between a bundler with an SxLP motif vs. ones without. The APC2 depletion experiments in neurons show microtubule dynamics and orientation of microtubules that remain dynamic post depletion of the APC2 protein, therefore ones that are unaffected by APC2. We have added more detailed explanations into the text of the manuscript related to these two experiments in order to clarify these differences and also clarified our conclusions in the Discussion section.

(ii) The C-terminus of APC2 also has an SxLP motif which binds to MT tips but the full length APC2 does not seem to bind to MT tips. Does endogenous APC2 exist in two forms – one binding to the MT tips and one binding to MT lattice? And can it switch from one form to another? If so, what would the switch be? Can the authors do single molecule motility assays with purified APC2 to nail down the specific effects of APC2 on MT dynamics?

These are interesting points raised by the reviewer. We would like to note that only the last half of the C-terminus (RFP-APC2-C2), which contains the SxLP motif, tracks growing microtubule tips. When the microtubule binding region is included in the construct (RFP-APC2-C), the lattice binding overrules the +TIP tracking capability. We never see full length RFP-APC2 clusters track growing microtubule tips, however we cannot rule out that it is possible. If it is possible, we imagine the switch from RFP-APC2 clusters tracking growing microtubule tips to binding the lattice would be controlled by a signaling cascade and perhaps phosphorylation of the protein. To our knowledge, purification of recombinant APC2 has not been done before. And while we agree that single molecule in vitro motility assay would shed further light on the mechanisms of function of APC2, we believe such a study is outside the scope of this paper.

6. The authors propose that APC2 clustering is driven by LC8 binding (Figure 7). The data in Figure 7E indicates that the C-terminus of APC2 recruits DYNLL2/LC8 to MTs. Is this true for full length APC2 as well?

We have done the requested experiment and added representative panels Figure 7e. Indeed, full length APC2 also recruits GFP-DYNLL2. Related quantifications can now be found in Figure 7f.

While the data in Figure 7A indicates that DYNLL2 co-localizes with APC2, what is the localization of DYNLL2 in neurons without any over-expression of APC2. Would it be cytoplasmic or punctae?

We have done the requested experiment and added a representative panel to Figure 7a showing that GFP-DYNLL2 is cytoplasmic in the control condition when co-expressed with RFP. Furthermore, in Cos7 cells in Figure 7e, GFP-DYNLL2 in the control condition co-expressed with RFP is also cytoplasmic.

The authors also go on to identify the specific region in the C-terminus that is required for the binding of LC8. The authors then express the mutated APC2 in neurons and conclude that LC8 binding is required for the proper localization of APC2. However, previous reports have shown that mutation in the C-terminus of APC2 acts as a loss-of-function. Hence, the mislocalization in this case could be due to this. Can the authors knock down endogenous LC8 and then observe

localization of APC2? This would eliminate the use of mutant constructs and will also strengthen their claim about LC8 binding determining the localization of APC2 in neurons.

Thank you for this suggestion, it raises an interesting point. We would first like to note that the mutation in the C-terminus that acts as a loss-of-function is a frame shift mutation, which results in a premature stop codon. This mutation was published by the Noda lab highlighting the need for a functional C-terminus in APC2 protein function. The mutation that we have made is a 3 amino acid substitution at the end of the C-terminus, which is not expected to render the C-terminus fully nonfunctional. Furthermore, truncated RFP-APC2-C(c2AAA) still efficiently binds microtubules (Figure 7e). While it is possible that the substitution results in unexpected folding discrepancies in the full length protein, the two mutations are fundamentally different. We followed reviewer's advice and added an experiment depleting endogenous DYNLL2 and examining localization of RFP-APC2. It is important to note that depletion of DYNLL2 had severe effects on neurons, therefore we ended up imaging neurons after 2 days of knockdown, instead of the 4 days used for APC2 knockdown. Representative panels of DYNLL2 depletion driven effect of RFP-APC2 localization in dendrites can be found in Supplementary Figure 4e.

***** Reviewer #3 *****

The study titled “APC controls dendrite development by promoting microtubule dynamics” is generally appropriate for nature communications and should be acceptable with significant revisions. The authors show a novel phenotype for APC2 by illustrating that APC2 depletion causes a significant reduction in the number of dendritic spines in cultured rat hippocampal neurons and fewer retrograde microtubule comets. The authors then find that expression of an RFP-tagged APC2 localizes to microtubules in dendrites, which depend on elements in the C-terminal portion of APC, as loss of the C-terminus mobilizes APC2 puncta and reduces colocalization with microtubules. More careful investigation of the C-terminus reveals varying requirements of certain C-terminal elements, with the distal portion of APC2 binding microtubule +ends, while an expanded S-domain is responsible for binding microtubule bundles. Furthermore, this patterning of APC2 along microtubules is dependent on a domain in APC2 that also interacts with the dynein complex member LC8. However, the authors make some claims that are not sufficiently supported and rely on reagents that are not entirely validated, and therefore several major issues remain to be addressed before a recommendation for publication can be made.

Thank you for finding our work appropriate for publication in Nature Communications pending significant revisions. We have carefully gone through the comments and suggestions listed below and made significant improvements to the manuscript that we hope will fully address your concerns. Our significant revision includes numerous additional experiments that address reviewer comments, improved data representation via addition of two movies and schematics, and amended text of the manuscript to reflect improvements in wording of data explanations and conclusions.

Major issues:

1. The sole use of RFP-APC2 overexpression constructs to make conclusions about APC2 localization limits the significance of these findings. Given that the localization and behavior of overexpressed and tagged constructs in relation to endogenous protein are not always reliable,

RFP-APC2 should be validated by showing that expression of RFP-APC is capable of rescuing APC2 knockdown in terms of dendrite morphology.

We completely agree with the reviewer and have done a rescue experiment to validate RFP-APC2. In Figure 7i we now show that RFP-APC2 expressed in APC2 depleted neurons is able to rescue dendrite morphology when compared to RFP expressed in control shRNA transfected neurons and RFP expressed in APC2 depleted neurons measured by Sholl analysis.

Additionally, localization of RFP-APC2 should be validated by comparing it to the localization of endogenous APC2.

Thank you for this suggestion. We have not stained for endogenous APC2 in the original manuscript, because the commercial antibodies we have acquired, including ones from published works, do not successfully recognize rat APC2 (via immunocytochemistry and Western Blots: the antibodies were unable to recognize correct sized band and/or depletion of APC2). Since all available antibodies are raised against the human peptides, we concluded that the antibodies must recognize human APC2 and not rat. In order to address this comment of the reviewer, we have now confirmed this by successfully staining our human RFP-APC2 expressed in rat neurons in Supplementary Figure 1b. To confirm that our expressed APC2 localizes similarly to endogenous, we used 2 of the commercial antibodies (one against the C-terminus and another against the middle of APC2) to stain human iPSC derived neurons and confirmed that endogenous APC2 localizes in clusters in MAP2 positive processes (Supplementary Figure 1b).

2. The title of this study is “APC2 controls dendrite development by promoting microtubule dynamics”. However, the direct connection between dendrite morphology and the observed changes in microtubule dynamics was not clearly established. To support this claim, the authors should confirm that the APC2 mutants, which alter the localization and interaction with microtubules, have some effect on dendrite morphology. By determining if a particular mutant is capable of rescuing the APC2 knockdown phenotype with regards to dendrite morphology, the authors could make the argument that association between APC2 and microtubules is necessary for proper dendrite development.

Thank you for suggesting these experiments. To support the connection between dendrite morphology and changes in microtubule dynamics we performed two rescue experiments. First, we fortified the microtubule dynamics rescue experiment in which proportions of dynamic microtubules in APC2 depleted neurons are rescued by the full length APC2, but not one lacking the C-terminus, by adding condition of RFP-APC2(cAAA) (Supplementary Figure 2a). Second, we performed a rescue experiment of dendritic morphology, showing that RFP-APC2 largely rescues the morphology of APC2 depleted neurons, but RFP-APC2- Δ C and RFP-APC2(cAAA) do not, as assessed by Sholl analysis (Figure 7i). Together, these experiments show that proper association between APC2 and microtubules is necessary for both proper dynamics of microtubules and proper dendrite morphology. We decided against assessing effect of APC2 mutants on dendrite morphology by overexpression, since multiple day overexpression is likely to result in non-biologically relevant phenotypes in non APC2 depleted conditions. Therefore, we focused on the rescue experiments to address this comment.

Specific/ minor points and experimental controls:

Figure 1

- The authors only quantify dendritic spines that possess both Homer and Bassoon staining and note a significant reduction in dendritic spine quantity with APC2 knockdown. This allows for the possibility that APC2 knockdown may be causing reduced levels of Homer and Bassoon resulting in number of dendritic spines being inappropriately counted. A control for Homer and Bassoon levels should be done for APC2 shRNA vs control shRNA.

We have added quantification of all protrusions that are not Homer and/or Bassoon positive into Figure 1h and i as “other protrusions”. We have also added quantification of Homer and Bassoon levels in spines of control vs. APC2 depleted neurons. Results of this quantification can be found in Figure 1 legend (Homer: control shRNA 6797 ± 470 fluor. units vs. APC2 shRNA 8360 ± 783 fluor. units, $p=0.125$, Mann-Whitney U test; Bassoon: control shRNA 16670 ± 923 fluor. units vs. APC2 shRNA 16441 ± 1348 fluor. units, $p=0.589$, Mann-Whitney U test).

Figure 2

- Since APC2-RFP is being overexpressed, it is important show quantifications of any effects in dendrite or dendritic spine morphology with this background.

The reviewer brings up a good point. We have reviewed all the data in the manuscript and came to the conclusion that overexpression effects of RFP-APC2 are difficult to interpret from the point of biological significance. Therefore, we retained experiments where RFP-APC2 is expressed in endogenous APC2-depleted neurons and also for the purposes of localization only. Consequently, we have removed experiments examining overexpression effects, namely old Supplementary Figure 2c, d. In regards to this point of the reviewer, our data in Figure 7i and Supplementary Figure 2a shows that RFP-APC2 expression in endogenous APC2-depleted cells rescues effects up to control levels, suggesting that RFP-APC2 has no aberrant effects when compared to endogenous protein.

- In Fig2D, the authors claim that the majority of RFP-APC2 clusters colocalize with tyrosinated microtubule bundles. This should be demonstrated with quantification especially since determination based on the provided images alone is not clear.

We agree with the reviewer that the colocalization determination cannot be made based on images in old Figure 2D alone. We also found that it is difficult to definitively say that APC2 binds to either acetylated or tyrosinated tubulin using regular confocal microscopy in dendrites. Therefore, we repeated this experiment and quantified colocalization of RFP-APC2 in cell bodies of neurons extracted prior to fixation with bundles of microtubules that were stained for tyrosinated or acetylated tubulin and found no significant difference in colocalization Manders' coefficient values (0.42 ± 0.03 for acetylated tubulin and 0.39 ± 0.03 for tyrosinated tubulin, $p=0.484$, Independent samples T-Test). We have added this data to Figure legend 2e and adjusted our conclusions in the manuscript relating to this figure panel (now Figure 2e) in the second paragraph of Results section “APC2 clusters localize to microtubule bundles in dendrites” and in the Discussion.

- A control confirming that nocodazole and SpvB treatment is sufficiently disrupting the cytoskeleton should be included.

We have now added a control confirming that nocodazole and SpvB treatments sufficiently disrupt the cytoskeleton into Supplemental Figure 1c.

Figure 3 and Sup Figure 2

- In Fig3 G,H, quantifications for retrograde comets are given as percents of total comets. This is not ideal since a difference in percent retrograde comets could be due to changes in the actual number of retrograde comets or due to changes in anterograde comets. Presenting the data as it is in Fig.2E would be more helpful.

Thank you for this suggestion. We have added quantification from old Figure 3G, H in numbers into new Figure 3g, h. Because dendrites do not have equal numbers of comets, the percent graph controls for variations in number of comets between dendrites. Therefore, we decided on keeping both calculations, since they both carry informative data.

- Comparing MT assembly velocity in samples with addition of control shRNA in FigS2B with addition of just RFP in FigS2D reveals a substantial difference in two samples that should be similar. Given this discrepancy, showing MT assembly velocity in a sample where neither control shRNA or RFP is added would be helpful.

In our response to Figure 2 comment, we describe reasons for removing half of this data from the manuscript (old Supplementary Figure 2C, D). Additionally, there was a discrepancy in our calculation of comet velocity, which we have now fixed (Supplementary Figure 2c). In our original calculation the frame rate was not adjusted appropriately to the 2 frames per second that the movies were taken at. Once we fixed the frame rate, the calculations readjusted to the reported rates of 0.1-0.2 $\mu\text{m}/\text{sec}$. The values are now comparable published data from Yau et al., 2016, Stepanova et al 2003 and Sweet et al., 2011. We apologize for this error.

Figure 4

- It is unclear in Fig4B what the quantification from Fig4C represent. Also, explicit indication should be given for what the dotted lines and arrows represent in B.

We have added schematic representation for what is calculated in Figure 4c and d. We also provide better explanations in the figure legend as requested. Figure 4b represents microtubule bundle sliding. In order to show an example of whole microtubule cytoskeleton translocation, we have now added a movie (Movie 1).

Figure 5 and Sup Figure 3

- The legend on the graph in Fig6B (Area of cell %) is unclear given the description in the figure legend (Representative image of area of microtubules...)

Our understanding is that the comment is in regards to Figure 5B. We have clarified the figure legend and changed the axis name.

- RFP-APC and RFP-APC2-C2 should be quantified with the others in Fig2B.

Our understanding is that the comment is in regards to Figure 5B. The calculation excludes RFP-APC2 and RFP-APC2-C2 because of the nature of their localization. RFP-APC2 is localized in clusters and is therefore difficult to judge efficiency of binding based on area of cell, furthermore it would not be comparable to the rest of the constructs since they are all either diffuse or microtubule bundlers. RFP-APC2-C2 is mobile since it tracks microtubule +Tips and also cannot be compared to the rest of the constructs, therefore is excluded from the graph. We have added a brief explanation of this into the manuscript into first paragraph of Results section "Functional domains of APC2 cytoskeleton interacting region".

- As mentioned by the authors, level of RFP-APC2 expression can have profound effects on APC2-RFP localization. A control that confirms similar levels of expression for all mutants shown in Fig5 should be included.

The control showing non-significantly different levels of expression for all constructs analyzed is now included in Supplementary Figure 3b.

- Colocalization quantifications between RFP-APC2 fragments and SiR-Actin should be included.

We agree with the reviewer. This quantification is most appropriate for the HeLa experiment in Supplementary Figure 3d, since Cos7 cells do not contain traditional actin stress fibers. We have now included calculation of colocalizations of RFP-APC2 fragments and SiR-Actin for the HeLa experiment in Supplementary Figure 3e.

Figure 6

- The authors state “These results suggest that RFP-APC2-C may indeed prefer freshly assembled dynamic microtubules and stabilize them upon binding, which in turn increases their acetylation level with time”. To conclude this, an experiment looking at the relative amounts of acetylated tubulin in samples with overexpressed RFP-APC-C and RFP-APC vs control needs to be done.

Thank you for this suggestion. We have added calculation of acetylation levels in cells expressing RFP, RFP-APC2 and RFP-APC2-C into Supplementary Figure 3f.

- It is unclear whether the kymographs in Fig6E represent pre or post rapalog treatment.

Thank you for catching this ambiguity. We have now clarified the figure and the legend.

- A key point argued in Fig6 is that the RFP-APC-C2 fragment is capable of catching growing microtubule plus ends and stabilizing them when tethered to the cell membrane. This point would be better supported if a clear way of quantifying the images in Fig6G in terms of immobilization of MTs was included. These quantifications should include comparisons of the ability of APC-C, APC-C1 and APC-C2 to immobilize MTs.

The quantification in Figure 6f shows the difference in immobilization of microtubules at the membrane between FRB-RFP-APC2-C, C1 and C2. We improved the axis title to clarify what is calculated and the figure legend. Furthermore, we included a movie to better visualize the difference in microtubule recruitment capability of FRB-RFP-APC2-C, C1 and C2 (Movie 2).

Figure 7

- The authors state “RFP-APC2 clusters in neurons significantly colocalize with GFP-DYNLL2 and to a lesser degree with GFP-DYNLL1”, but no quantifications are shown.

Thank you for catching this incompleteness. We have now included a quantification in Supplementary Figure 4d.

- Fig7G should include quantifications of colocalizations between RFP-APC2 and GFP-DYNLL2.

Our understanding is that the comment is in regards to Figure 7E. We have now added this condition into Figure 7e and quantification into Figure 7f.

- To establish that DYNLL2 is responsible for proper localization of RFP-APC2, DYNLL2 should be depleted and localization of RFP-APC2 should be assessed.

Thank you for this suggestion. We have done the requested experiment and added representative panels to Supplementary Figure 4e. It is important to note that depletion of DYNLL2 had severe effects on neurons, therefore we ended up imaging neurons after 2 days of knockdown, instead of the 4 days used for APC2 knockdown.

***** Reviewer #4 *****

This is an exciting manuscript, executed beautifully, and presented magnificently. The question addressed here is important to neuroscientists interested in how neurons regulate the various details of their microtubule arrays. The question might be a tad narrow for a general audience, but the work is so masterfully done that I believe cell biologists will find insight relevant to many cell types in this work. The issue is how the minus-end-out microtubules in the vertebrate dendrite have an overall different level of stability than the plus-end-out microtubules, with the conclusion being that APC2 clusters keep the plus ends of the minus-end-out-microtubules more dynamic. There's a great deal of data in the paper, certainly no need for more, with a number of different molecular factors and microtubule behaviors studied.

We are very grateful for your comments and for finding our work exciting and well executed. In our revised manuscript we have added several new experiments, improvements of the text, and two movies.

REVIEWERS' COMMENTS:

Reviewer #2 (Remarks to the Author):

The authors addressed all the points raised by the reviewers and the manuscript is much easier to follow now with the several text changes made by the authors. I recommend publication.

Reviewer #3 (Remarks to the Author):

The authors have revised the manuscript substantially with many new sets of new data and important clarifications/corrections. The new version addresses the majority of issues raised and thus should be entirely appropriate for publication in this journal.